# *Staple:* Towards Reliable Problem Solving with Large Language Models via Plan Optimization and Tree Search

## Abstract

Large language models (LLMs) exhibit the ability to perform step-by-step reasoning when tackling complex problems across various tasks. To improve the reliability of multi-step reasoning and mitigate potential hallucinations, sophisticated prompting techniques have been developed to provide instructions on *what to do* at each step, offering reasoning guidance before addressing specific questions. However, this additional prompting can increase time and token consumption without guaranteeing effectiveness. In response, this paper proposes *Staple*, a novel plan retrieval augmented reasoning framework that utilizes *offline plan optimization*. This approach involves constructing a plan database of general-purpose reasoning instructions. Subsequently, *online plan searching* facilitates the direct retrieval of optimal and effective step-by-step plans from the database when addressing new questions, serving as guidance for LLMs to derive correct answers. The *offline* stage uses LLMs to self-generate and optimize plans, storing them as tree structures via Monte Carlo Tree Search (MCTS) to form the plan database. Extensive experiments on mathematical and multi-task problems show that *Staple* achieves competitive problem-solving rates while minimizing token usage and interactions. Importantly, the plan trees in the database are human-interpretable, revealing the prioritization of various plan combinations for a given task. In addition, the plan database can be reused, updated, and expanded by users for a wider range of applications.

## 1 Introduction

Large language models (LLMs), trained via autoregressive text token prediction, exhibit the capacity for multi-step reasoning to solve complex tasks. When presented with a prompt, these models can be directed to approach problems step-by-step (Kojima et al., 2022), with each step referred to as a *thought* and representing a solution to a simplified subproblem. However, due to LLMs' susceptibility to hallucinations, incorrect or invalid thoughts are often produced (Jiang et al., 2023; Ji et al., 2023), highlighting a crucial need to enhance the reliability of their problem-solving abilities.

Existing literature addresses this objective by prompting LLMs with step-by-step instructions. Existing approaches relying on hand-crafted demonstrations Wei et al. (2022); Zhou et al. (2023a); Fu et al. (2023) or domain-specific knowledge databases Sun et al. (2024); Luo et al. (2024) may have limited applicability. Recent proposals allow LLMs to autonomously generate error analyses Madaan et al. (2023); Chen et al. (2024); Miao et al. (2024); Chen & Li (2024), plans Wang et al. (2023); Yu et al. (2024); Zheng et al. (2024b) or premises Ling et al. (2023) as instructions while tackling individual questions. However, beyond consuming token and time resources, engaging LLMs in instruction generation remains susceptible to hallucinations Valmeekam et al. (2023), suggesting that the assumption of LLMs producing accurate guidance may be unfounded. Consequently, we pose the question: "How can we autonomously generate reliable instructions for each step during multi-step problem-solving while minimizing token and time costs?"

This research challenge remains unresolved, primarily because formulating an effective instruction is as challenging as selecting a move in The Game of Go Müller (2002). Given existing thoughts, it remains unclear how LLMs can explore or exploit an instruction that yields an improved subsequent thought, particularly when the solution requires numerous steps or is unknown. Furthermore, in contrast to the finite sets of moves and board positions analyzed by AlphaGo Silver et al. (2016), the thoughts and instructions generated by LLMs are unbounded, potentially infinite, and susceptible to errors, especially due to the diversity of questions and the possibility of hallucinations. In contrast, human next-step reasoning is guided by a *plan*, which serves as a general-purpose instruction presenting specific logic, such as "Use variables to represent unknown quantities" or a theorem. Plans necessary for addressing a given task can be standardized and limited in scope, allowing humans to first acquire experience on plan priorities, and subsequently select effective ones for new, related questions.

In this paper, we propose *Staple*, a new plan retrieval augmented reasoning (RAR) framework designed to incorporate both *offline plan optimization* and *online plan searching*. During the offline stage, *Staple* optimizes plans based on a task-specific samples. By leveraging LLMs for step-wise plan proposal and reasoning on labeled questions, *Staple* expands the plan space as a tree structure using Monte Carlo Tree Search (MCTS) Coulom (2006). Beginning with an empty plan tree, *Staple* utilizes LLMs to either expand the plan (node) or select high-priority options to simulate multi-step reasoning towards a solution, subsequently updating plan sequence priorities during backpropagation. Notably, as plan (node) selection relies on the previous thought (state), *Staple* allows each node to store all such thoughts. Thus, by exploiting the clustering nature of embeddings for similar thoughts, and by incorporating the distance between current and stored thoughts into priority scores, we are able to avoid searching in an infinite state space. Upon completion of offline training, *Staple* generates a plan database in which each plan tree corresponds to a specific task, such as a particular domain of mathematical problems. During the online stage, when presented with a new question, *Staple* identifies high-priority plans from the relevant plan tree to guide LLMs towards reliable multi-step reasoning.

With *Staple*, we make a number of important contributions in this paper. First, it provides a generalized framework that allows LLMs to autonomously explore and optimize task-specific plans without human intervention. Second, *Staple* uncovers numerous plan combinations and their associated priorities to address task-related questions. Third, it is inherently effective, resource-efficient, and user-friendly, as the reliability of LLM reasoning is ensured by effective plans retrieved from *Staple*'s plan database tree, and the task-specific plan trees can be reused, updated, and extended. Finally, our comprehensive set of experiments on mathematical and multi-task problems corroborate these advantages and demonstrate *Staple*'s competitive solving rates.

## 2 RELATED WORK

Pre-trained large language models (LLMs), such as GPT-4o OpenAI (2023) and Llama 3.2 Touvron et al. (2023), possess the capability of **multi-step reasoning**, wherein problems are solved incrementally Kojima et al. (2022) with intermediate steps manifested as thoughts generated by LLMs. Nevertheless, LLMs often confidently produce incorrect or invalid thoughts due to hallucinations Jiang et al. (2023); Ji et al. (2023); Zheng et al. (2024a). Consequently, prompting LLMs with instructions on *what to do* at each step is crucial for dependable problem-solving. Chain of Thoughts Wei et al. (2022) and subsequent works Wang et al. (2022); Zhou et al. (2023a); Fu et al. (2023); Chen et al. (2023a); Weng et al. (2023) encourage LLMs to emulate human-made demonstrations. Notably, Auto-CoT Zhang et al. (2023b) samples examples from the training set based on question clustering. Furthermore, without designing specific prompts, existing approaches like ToT Yao et al. (2023) and GoT Besta et al. (2023) allow LLMs to backtrack and self-evaluate during the reasoning process. Recent research Madaan et al. (2023); Chen et al. (2024); Chen & Li (2024); Miao et al. (2024) utilizes LLMs to perform self-reflection, thereby collecting error analyses in the prompt to guide LLMs towards accurate thought generation. Rather than becoming ensnared in these resource-intensive approaches, this paper initially learns the

plan, an effective high-level concept of what to do in the subsequent step, to prompt LLMs with appropriate plans for reliable problem-solving.

Similar to our plan-based problem solving, there also exists literature on **augmenting LLMs with step-wise planning**. Existing approaches Wang et al. (2023); Zhang et al. (2023a); Ling et al. (2023); Wang et al. (2023) relying on the divide and conquer mechanism break down a task into smaller, manageable sub-problems based on plans derived from LLMs. For example, having LLMs perform analogical reasoning Yu et al. (2024); Zheng et al. (2024b) to acquire high-level strategies for facilitating multi-step reasoning proves more effective. Specifically, to enhance problem reasoning, STEP-BACK prompting Zheng et al. (2024b) initially learns a generic strategy from an abstracted version of the problem. Moreover, combining LLMs with the heuristic algorithm, Monte Carlo Tree Search (MCTS) Coulom (2006) demonstrates competitive planning performance Zhao et al. (2023). However, LLMs struggle to generate plans that are guaranteed to be correct Valmeekam et al. (2023). In contrast, our offline plan optimization initially constructs a plan database containing effective plans for tasks. Subsequently, online plan searching retrieves the most useful plans to guide LLMs towards reliable problem-solving.

The prompting framework of optimizing first and applying afterward in *Staple* closely resembles **prompt optimization** Zhou et al. (2023b); Guo et al. (2024); Wang et al. (2024). In this approach, LLMs optimize a simple prompt based on labeled questions for a single task, thereby producing high-quality instructions that benefit the addressing of new questions. Notably, PromptAgent Wang et al. (2024) views prompt optimization as a strategic planning problem, utilizing Monte Carlo tree search (MCTS) and self-reflection Weng et al. (2023) of LLMs to explore expert-level prompts. MoT Li & Qiu (2023) introduces a method where LLMs generate and store useful thoughts, which are extracted as instructions to guide them in solving new questions. Our work advances significantly by optimizing step-wise plans rather than improving a single prompt, which can guide LLMs towards reliable multi-step problem-solving. Specifically, post-optimization, *Staple* generates a plan tree for each task, illustrating which plans can be employed in each step and which are superior. We aim for *Staple* and the database containing plan trees to inspire broader applications of LLMs.

## 3 Methodology: First Plan Optimization, Then Retrieval

### 3.1 Overview

Given a question $q$ from the dataset $D$ of a task $\mathcal{T}$, the multi-step problem-solving process with an LLM $f_\theta$ involves guiding this pre-trained model with an input prompt $\mathbb{I}(\cdot)$ to perform multi-step reasoning toward a solution $y$. In this process, the intermediate steps are represented as thoughts $\mathbf{Z} : \boldsymbol{z}_{0\ldots T} = [z_0, z_1, \ldots, z_n \ldots, z_T]$ generated by LLMs, where $z_0$ contains $q$, $z_n$ is the $n$-th thought, and $z_T := y$ is the final solution thought. We specifically focus on a multi-interaction approach Chen et al. (2024); Chen & Li (2024), in which each interaction with the LLM produces one subsequent thought, formulated as $z_n \sim f_\theta \left(z | \mathbb{I}_G \left(\boldsymbol{z}_{0\ldots n-1}\right)\right)$. To mitigate hallucinations in $z_n$, it is essential to consistently include an effective instruction $\psi_n$ that presents *what to do* as reasoning guidance in $\mathbb{I}_G$, resulting in $\mathbb{I}_G \left(\boldsymbol{z}_{0\ldots n-1}, \boldsymbol{\psi}_{1\ldots n}\right)$, where $\boldsymbol{\psi}_{1\ldots n} = [\psi_1, \ldots, \psi_n]$ represent the instructions for $n$ steps.

In this context, our primary objective is to autonomously produce a sequence of reliable instructions, thus eventually leading to the reasoning chain $z_0 \xrightarrow{\psi_1} z_1 \ldots \xrightarrow{\psi_n} z_n \xrightarrow{\psi_{n+1}} \ldots z_T$, which gives a correct solution while minimizing both token and time costs.

In this paper, drawing inspiration from human-like reasoning — where high-level principles for addressing a task's questions can be fixed and limited in number — we introduce the concept of a *plan* 3.1. With a plan and given a task, we can reduce the search space by transforming the instruction space into a plan space, resulting in a more tractable problem. This approach derives from the intuition that, due to the clustering nature of semantically similar texts, *thoughts with more closely aligned embedding vectors are more likely to follow the same plan.*

**Definition 3.1** (*plan*). A plan is a high-level, question-agnostic principle that aids in deducing a single logical reasoning step towards addressing a specific task. When provided

with a prompt containing this general-purpose reasoning instruction, LLMs are guided to employ a particular logical approach to generate a valid thought leading to the correct solution.

Ultimately, we can simplify the aforementioned objective by implementing a two-stage approach: initially, we explore and optimize plans for a specific task, enabling LLMs to subsequently search for the most effective plans step-by-step when addressing related downstream questions. Consequently, our proposed plan retrieval augmented reasoning (RAR) framework, *Staple*, encompasses both of these stages, henceforth referred to as *offline plan optimization* and *online plan searching*.

### 3.2 OFFLINE PLAN OPTIMIZATION

Based on the labeled questions from the training dataset of a given task, the plan optimization problem is formulated as $\mathbf{P}^* = \arg\min_{\mathbf{P}} \sum_{(q,y) \in D_r} \mathbf{1}\left(y, f_\theta\left(q, \mathbf{P}\right)\right)$, where $D_r$ denotes labeled samples and we aim to optimize a plan selection method $\mathbf{P}$ to consistently yield correct solutions, as measured by an indicator function $\mathbf{1}$. As discussed in Section 3.1, this strategic planning problem is naturally addressed by Monte Carlo Tree Search (MCTS) Coulom (2006), which structures $\mathbf{P}$ as a decision tree. Leveraging the inherent capabilities of LLMs, we integrate $f_\theta$ to develop an effective algorithm as follows.

**Plan tree**. A task's plan tree $\mathbf{P}$ contains a Plan Set $\mathbf{\Psi}$ and Plan Thought $\mathbf{\Upsilon}$, with each node corresponding to one plan. The plan (node) set $\mathbf{\Psi}_n$ of the $n$-th level presents plan candidates for the $n$-th thought generation. Let $\psi_{ni}$ denote the plan with index $i$ of the $n$-th level; its parent node (plan) and child nodes are denoted as $\psi_{n-1j}^i$ and $C\left(\psi_{ni}\right)$, respectively. For $\psi_{ni}$, its corresponding $\mathbf{\Upsilon}_{ni}$ stores any thought $z_{n-1}$ that leads to this plan. For instance, in $\mathbf{\Psi}_1$ of the 1-st level, the $\mathbf{\Upsilon}_{1i}$ of the $i$-index plan contains all questions $z_0 =: q$ that embrace $\psi_{1i}$ plan as guidance when generating the first reasoning step $z_1$. See Fig. 6 and Section D of the Appendix for more insights.

**Plan generation**. Any plan $\psi_{ni}$ in the $n$-th level is generated only by using the LLM to *summarize* one from the generated thought $z_n$, which is formulated as $\psi_n \leftarrow f_\theta\left(\psi | \mathbb{I}_S\left(\boldsymbol{z}_{0,...,n}, \boldsymbol{\psi}_{0,...,n-1}\right)\right)$, where $\mathbb{I}_S$ is the plan summarization prompt and $\boldsymbol{\psi}_{0,...,n-1} = \left[\psi_{00}, ..., \psi_{(n-1)*}\right]$ contains a sequence of plans selected in the previous $n-1$ steps, with $\psi_{00}$ as the root and $*$ as a general denotation representing a selected plan. As the next step of existing thoughts $\boldsymbol{z}_{0...n-1}$, $z_n$ generated by LLMs is essentially a trial of reasoning and thus implicitly follows the LLM's internal logic in this reasoning chain $\boldsymbol{z}_{0...n}$. A plan summarized from this thought maintains such logic and can thus be used as guidance for the $n$-th thought generation for similar questions.

**Plan exploration**. Since $\mathbf{P}$ is initially empty, there should be a trade-off between exploration (generating new plans) and exploitation (using existing plans). Thus, we set the rule: for each plan (node) $\psi_{ni}$, the probability of exploration $p_{\psi_{ni}}$ is a sigmoid function $1 / \left(1 + e^{0.2(|C(\psi_{ni})| - M)}\right)$, where $M$ is a constant value. When the probability is larger than 0.5, we use the LLM to generate a thought by excluding existing plans in $C\left(\psi_{ni}\right)$, which is formulated as $z_{n+1} \leftarrow f_\theta\left(z_{n+1} | \mathbb{I}_E\left(\boldsymbol{z}_{0,...,n}, C\left(\psi_{ni}\right)\right)\right)$, where $\mathbb{I}_E$ is the plan exclusion prompt. Then, by summarizing from this thought, we generate a new plan and add it to $C\left(\psi_{ni}\right)$.

**Plan comparison**. Once a plan $\psi$ is summarized from a thought, the LLM with a plan comparison prompt $\mathbb{I}_C\left(\psi, \boldsymbol{\psi}\right)$ is used to check whether $\psi$ exists in a set $\boldsymbol{\psi}$.

**Thought comparison**. Since semantically similar thoughts have closer text embeddings, a generated thought $z_n$ is compared with $\mathbf{\Upsilon}_{n+1}$. That is, for the $i$-index plan (node) of the $n$-th level, the embedding distance $d\left(z_n, \mathbf{\Upsilon}_{(n+1)ik}\right)$ shows the similarity between the thought $z_n$ and a stored thought with index $k$ of the node. Thus, we set $K\left(z_n, \mathbf{\Upsilon}_{(n+1)i}\right)$ as the number of $K$ neighbors of $z_n$.

**Reward assignment**. Our *Staple* assigns the reward to each thought of the plan, meaning that $r\left(\mathbf{\Upsilon}_{nik}\right)$ is the reward of the $k$-index thought of the $i$-index plan of the $n$-th level. Additionally, $r$ contains two parts: $r_w, r_{llm} \in [0, 1]$. $r_w$ is the indicator of win, noting whether selecting the plan $\psi_{ni}$ for the next thought generation for $\mathbf{\Upsilon}_{nik}$ leads to the final

---

**Algorithm 1:** Plan Optimization in *Staple*

---

**Input:** LLM $f_{\theta}$, Plan Tree $\mathbf{P} = \{\boldsymbol{\Psi}, \boldsymbol{\Upsilon}\}$, Question $(q, y)$.
**Output:** Optimized $\mathbf{P}$.

---

**1** $\boldsymbol{z}_0 = [z_0 := q]$, $\boldsymbol{\psi}_0 = [\psi_{00}]$, $n = 0$ //`Start the multi-step reasoning`
**2** **while** *not $\boldsymbol{z}_n$ reaches solution* **do**
**3**    **if** $p_{\psi_{n*}} >= 0.5$                                    ▷ `Plan exploration`
**4**       $z_{n+1} \leftarrow f_{\theta}\left(z | \mathbb{I}_E\left(\boldsymbol{z}_{0,\dots,n}, C\left(\psi_{n*}\right)\right)\right)$ //`Generate next thought excluding` $C\left(\psi_{n*}\right)$
**5**    **else**
**6**       $z_{n+1} \sim f_{\theta}\left(z | \mathbb{I}_{G'}\left(\boldsymbol{z}_{0\dots n}, \boldsymbol{\psi}_n\right)\right)$ //`Generate next thought normally`
**7**    **end**
**8**    $\psi_{n+1} \leftarrow f_{\theta}\left(\psi | \mathbb{I}_S\left(\boldsymbol{z}_{0,\dots,n+1}, \boldsymbol{\psi}_{0,\dots,n}\right)\right)$                    ▷ `Plan generation`
**9**    **if** $\psi_{n+1} \in C\left(\psi_{n*}\right)$                                  ▷ `Plan comparison`
**10**       //`Selection of MCTS:`                           ▷ `Best plan first`
**11**       $\psi_{(n+1)*} = \psi_{(n+1)i^*}$ where $i^* = \arg\max_i \left\{ V\left(\psi_{(n+1)i}, \boldsymbol{\Upsilon}_{(n+1)i}, z_n\right), i \in C\left(\psi_{n*}\right)\right\}$
**12**       $z_{n+1} \leftarrow f_{\theta}\left(z | \mathbb{I}_G\left(\boldsymbol{z}_{0\dots n}, [\psi_{00}, \psi_{1*}, \dots, \psi_{n*}, \psi_{(n+1)*}]\right)\right)$ //`Generate next thought`
**13**       $v_{llm} = f_{\theta}\left(v | \mathbb{I}_A\left(\boldsymbol{z}_{0,\dots n}, \psi_{(n+1)*}, z_{n+1}\right)\right)$ //`Plan assessment`
**14**       $\boldsymbol{\Upsilon}_{(n+1)i^*} \leftarrow \left[\boldsymbol{\Upsilon}_{(n+1)i^*}, z_n\right]$, $\boldsymbol{\psi}_{n+1} \leftarrow \left[\boldsymbol{\psi}_n, \psi_{(n+1)*}\right]$, $\boldsymbol{z}_{n+1} \leftarrow \left[\boldsymbol{z}_n, z_{n+1}\right]$, $n \leftarrow n+1$
**15**    **else**
**16**       //`Expansion of MCTS:`                          ▷ `Create new plan`
**17**       $v_{llm} = f_{\theta}\left(v | \mathbb{I}_A\left(\boldsymbol{z}_{0,\dots n}, \psi_{n+1}\right)\right)$                    ▷ `Plan assessment`
**18**       $C\left(\psi_{n*}\right) = C\left(\psi_{n*}\right) \cup \psi_{n+1}$, $\boldsymbol{\Upsilon}_{(n+1)|C(\psi_{n*})|} \leftarrow \left[\boldsymbol{\Upsilon}_{(n+1)|C(\psi_{n*})|}, z_n\right]$
**19**       $\psi_{(n+1)*} = \psi_{n+1}$, $\boldsymbol{\psi}_{n+1} \leftarrow \left[\boldsymbol{\psi}_n, \psi_{(n+1)*}\right]$, $\boldsymbol{z}_{n+1} \leftarrow \left[\boldsymbol{z}_n, z_{n+1}\right]$, $n \leftarrow n+1$
**20**       **break**
**21**    **end**
**22** **end**
**23** //`Simulation/RollOut of MCTS:`                   ▷ `Reason toward solution`
**24** $m \leftarrow n$
**25** **while** *not $\boldsymbol{z}_m$ reaches solution* **do**
**26**    $\psi_{(m+1)*} = \psi_{(m+1)i}$ where random $i \in C\left(\psi_{m*}\right)$ //`Random plan selection`
**27**    $z_{m+1} \leftarrow f_{\theta}\left(z | \mathbb{I}_G\left(\boldsymbol{z}_{0\dots m}, \boldsymbol{\psi}_m, \psi_{(m+1)*}\right)\right)$, $\boldsymbol{z}_{m+1} \leftarrow \left[\boldsymbol{z}_m, z_{m+1}\right]$, $m \leftarrow m+1$
**28** **end**

---

**29** ▷ `Backpropagate` $\mathbf{1}(y, \boldsymbol{z}_m)$ `to visited nodes based on` **`Reward assignment.`**

correct solution. The $r_{llm}$ indicates the LLM's evaluation score for selecting the $i$-index plan to guide the next thought generation after reaching the thought $\boldsymbol{\Upsilon}_{nik}$. With the plan assessment prompt $\mathbb{I}_A$, $r_{llm}$ is obtained from $f_{\theta}\left(v | \mathbb{I}_A\left(\boldsymbol{z}_{0,\dots n-1}, z_n, \psi_{ni}\right)\right)$.

**Value function**. When reaching a thought $z_n$ guided by the plan $\psi_{nj}$, the priority score $V(\cdot)$ of selecting a next plan $\psi_{(n+1)i}$ is $\lambda \overline{r}_{llm} + (1 - \lambda)\overline{r}_w + 1 / \overline{\mu}\left(K\left(z_n, \boldsymbol{\Upsilon}_{(n+1)i}\right)\right)$, where $i \in C\left(\psi_{nj}\right)$, $\overline{r}_{llm}$ and $\overline{r}_w$ compute the average $r_{llm}$ and win rate of $K$ neighbors, respectively, i.e., $K\left(z_n, \boldsymbol{\Upsilon}_{(n+1)i}\right)$, and $\overline{\mu}(\cdot)$ computes the average distance of text embeddings.

With these designs, we can complete plan optimization by allowing the LLM to self-reason through each question in the dataset $D_r$ for $E$ epochs. In multi-step reasoning for each question, one MCTS iteration is performed to explore and optimize the plan tree, the process of which is shown by Algorithm 1, Fig. 5 and Fig. 6.

### 3.3 ONLINE PLAN SEARCHING

After the offline stage, *Staple* acquires a plan database comprising numerous plan trees. Each tree originates from a specific task and is tagged with a category name, such as "Math/Algebra" or "Math/Number Theory" from the `MATH` dataset. Therefore, *Staple* ensures reliable problem-solving by retrieving the highest-priority plans from the tree as reasoning guidance.

When presented with a question, *Staple* initially retrieves the relevant plan tree by matching the question's category with the corresponding tag. Following this, *Staple* searches for plan

combinations within the tree to guide the LLMs towards reliable multi-step reasoning for answering the question. Two methods of plan searching are employed. The first method, termed **Direct**, selects the plan with the highest $r$ at each level, thereby obtaining a sequence of plans. Using these plans in the prompt $\mathbb{I}_G$, the LLM answers the question in a single interaction. The second method, **Adaptive**, dynamically selects plans during the reasoning process. Specifically, once a thought $z_n$ is generated, the plan with the highest $V(\cdot)$ is chosen to prompt the LLM to produce the next thought $z_{n+1}$. Consequently, plans are adaptively searched until reaching the tree's leaf, which represents a solution. Inspired by BoT Chen et al. (2024) and TR Chen & Li (2024), we also investigate the inclusion of plans with the lowest value in the prompt as negative examples, guiding the LLM to avoid incorrect reasoning. This approach is referred to as **D-Contrastive**. For more concrete examples of how **Adaptive** tackles the question, refer to Fig. 7 and Fig. 8.

## 4 EXPERIMENTS

**Datasets**. We perform experiments on two categories of tasks using `AQUA-RAT`$_{97467/800/254}$ Ling et al. (2017), `MATH`$_{7500/700/900}$ Hendrycks et al. (2021), and `TheoremQA`$_{800/800/800}$ Chen et al. (2023b) (no-visual) datasets. The numerical subscripts $a/b/c$ denote the total number of samples, training samples, and testing samples, respectively.

**Large language models**. We employ GPT-3.5-turbo (gpt-3.5-turbo-16k-0613), GPT-4 (gpt-4-0613) OpenAI (2023), and Llama 2 Touvron et al. (2023), which includes Llama 2-13b (Llama-2-13b-chat-hf) and Llama 2-70b (Llama-2-70b-chat-hf), where 1b represents one billion parameters. For LLMs using *Staple*, the Temperature/Top_P settings for $\mathbb{I}'_G$, $\mathbb{I}_G$, $\mathbb{I}_S$, $\mathbb{I}_E$, $\mathbb{I}_C$, and $\mathbb{I}_A$ are 0.6, 0.4, 0.4, 0.2, and 0.2, selected based on empirical intuition. Across all experiments, the offline stage consists of 10 epochs, with $\lambda$ set to 0.3. Additionally, $M$ for plan exploration is fixed at 5 and the text-embedding-3-small is used as the embedder while the cosine distance is used for text distance measurement. For comprehensive details and insightful experiments, refer to Section A in the Appendix.

**Competitors**. Baseline methods include Zeroshot, Zeroshot-CoT Kojima et al. (2022), Chain-of-thought (CoT) Wei et al. (2022), and Complex CoT Fu et al. (2023) (C-CoT), each consistently using 8 shots. The state-of-the-art competitors are Tree of Thoughts (ToT) Yao et al. (2023), Cumulative Reasoning (CR) Zhang et al. (2023a), Boosting of Thoughts (BoT) Chen et al. (2024), Thought Rollback (TR) Chen & Li (2024), and STEP-BACK Zheng et al. (2024b), with their optimal settings applied. ToT Reasoning utilizes ToT with a breadth limit of 6 following the best first search (BFS). To facilitate the ablation study, we incorporate multi-interaction reasoning (Chain Reasoning) without a plan, *Staple*-Direct, and *Staple*-Adaptive. Notably, we implement *Staple* with the ensemble method (Ensemble-$C$), wherein #$C$ plan chains are extracted from the plan tree for reasoning with LLMs, and the final solution is determined by majority voting of their individual solutions.

**Metrics**. All experiments report the Solve Rate (%), calculated by comparing the solution following "The solution is" in $z_{0...T}$ with the ground truth. Furthermore, we record the number of interactions and tokens necessary to solve a single problem using the LLM. See Section A of the Appendix for Reproducibility.

### 4.1 OVERALL PERFORMANCE

**Plan database**. Following the offline optimization stage in *Staple* with GPT-4, we acquire three plan databases corresponding to `AQUA-RAT`, `MATH`, and `TheoremQA`, with the number of internal plan trees being 1, 7, and $14 + 5 + 14 + 6$, respectively. The notation $14 + 5 + 14 + 6$ represents the number of categories across four fields: $MATH$, $EECS$, $Physics$, and $Finance$ in the `TheoremQA` dataset. Each plan tree is tagged with a category name, allowing for retrieval upon matching the category of a new question. For example, the plan tree tagged with $EECS/InformationTheory$ contains plans with optimized priorities for questions within this knowledge domain. Consequently, the plan tree is inherently reusable as the general instruction for questions from the same domain follows consistent logic. The number of nodes (plans) in these three plan databases is 2001, 2283, and 2435, respectively, indicating

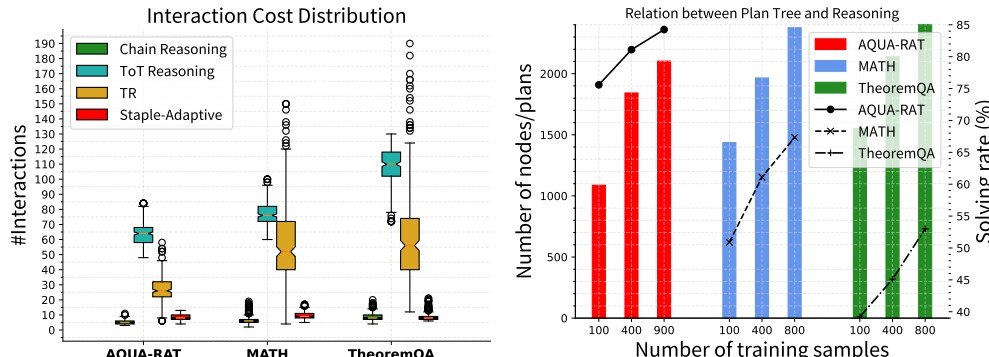

Figure 1: *Staple*'s effectiveness in reducing interaction costs and optimizing plan trees. The interaction count is determined by employing the *Staple*-Adaptive retrieval method with GPT-4 to address questions from each dataset. The right subfigure illustrates that as the number of training samples increases, *Staple* constructs a plan tree with more nodes/plans, resulting in improved solving rates.

that *Staple* synthesizes and optimizes numerous plan trials during optimization. Moreover, *Staple* automatically obtains a plan to guide LLMs in self-checking the reasoning step for revisions, as illustrated in Fig. 8 (red color) and Section E.1 (Bold plan).

**Effectiveness**. As demonstrated by Fig. 1 and Table 1, *Staple* achieves competitive solving rates with minimal token cost and interaction numbers (#Interactions). Our maximum cost *Staple*-Adaptive method reduces #Interactions by factors of 5, 7, and 9, respectively, compared to TR Chen & Li (2024). In comparison to ToT reasoning Yao et al. (2023), which requires approximately 100 #Interactions for challenging `MATH` and `Theorem` tasks, the cost-saving benefits of *Staple* are even more substantial. Regarding token cost for reasoning on the three datasets, *Staple*-Direct is 72, 38, and 31 times less than TR Chen & Li (2024) and Chen et al. (2024), while the values for *Staple*-Adaptive are 9, 8, and 7, respectively. By reducing interaction count and prompt token cost to levels comparable with zero-shot multi-round reasoning (Chain Reasoning), our method enhances usability in real-world applications. Notably, the solving rate of *Staple*-Adaptive remains the second-best across all three challenging datasets. *Staple*-Direct directly achieves higher solving rates than ToT Reasoning Yao et al. (2023) and C-CoT Fu et al. (2023). Compared with the state-of-the-art competitor, TR Chen & Li (2024), *Staple* reduce the interaction and token cost by approximately 10 times, with only a slight decrease in the solving rate.

**Flexibility**. As the plan tree encapsulates the general and high-level plans to address a specific problem, each path of the tree represents a distinct reasoning logic for tackling the question. The plan database exhibits high flexibility, as many existing reasoning algorithms can be integrated into or achieved by the plan tree. In the lower block of Table 1, we specifically incorporate ensemble Wang et al. (2022) and contrastive Chen et al. (2024); Chen & Li (2024) approaches at the retrieval stage in *Staple*. Contrary to our expectations, the solving rate of D-contrastive does not demonstrate a clear improvement. When 10 paths with lower priorities from the plan tree are included in the prompt, the value decreases by 0.75 compared to that of 5 paths. Similarly, for the ensemble method, the solving rate is competitive with *Staple*-Adaptive only when 80 paths are involved in solution voting. This result suggests that the plan tree is already optimized, as each node/plan incorporates historical thoughts with their corresponding $r_{llm}$ and $r_w$. The sole requirement is to adaptively identify the best plan for each step from numerous candidates in each layer of the tree.

**Reliability**. Accurately solving a given question while avoiding hallucinations and presenting human-explainable intermediate reasoning logic is the principal advantage of *Staple*. Firstly, as shown in Table 1, *Staple* is confirmed to achieve the second-best solving rate across all three most challenging tasks. This is attributed to the fact that with a sequence of effective plans, LLMs are guided to perform reasoning, thus significantly reducing the frequency of hallucinations. Secondly, during the online retrieval stage, for each thought, we consistently select the plan based on the principle that after using this plan, similar thoughts achieve the

Table 1: Comparing the *Staple* with other baseline methods by performing GPT-4 on the test sets of `AQUA-RAT`, `MATH` and `TheoremQA`. The metrics used here are solving rate (SR %), and the number of prompt tokens (#Tokens). The unit of the quantity of #Tokens is $1K$, meaning 1000 per unit. We show the mean $\pm$ standard deviation.

| Methods | AQUA-RAT | | MATH | | TheoremQA | |
|---|---|---|---|---|---|---|
| | SR | #Tokens | SR | #Tokens | SR | #Tokens |
| ZeroShot | 50.4 | 0 | 42.2 | 0 | - | 0 |
| Zeroshot-CoT Kojima et al. (2022) | 73.2 | $0.09 \pm 0.02$ | 44.7 | $0.1 \pm 0.03$ | 40.8 | $0.12 \pm 0.04$ |
| C-CoT Wei et al. (2022) | 75.2 | $3.3 \pm 1.2$ | 48.93 | $4.6 \pm 1.9$ | - | - |
| PHP+C-CoT | 79.9 | $7.9 \pm 2.3$ | 53.9 | $11.34 \pm 3.1$ | - | - |
| Chain Reasoning | 74.41 | $2.1 \pm 1.1$ | 45.4 | $2.7 \pm 1.2$ | 36.5 | $4.6 \pm 7.6$ |
| ToT Reasoning Yao et al. (2023) | 76.38 | $6.8 \pm 2.7$ | 48 | $9.9 \pm 3.5$ | 38.38 | $11.9 \pm 8.9$ |
| BoT Chen et al. (2024) | 81.4 | $42.4 \pm 33$ | 62.9 | $51.3 \pm 41$ | - | - |
| TR Chen & Li (2024) | 79.97 | $38.4 \pm 29.8$ | 71.89 | $46.9 \pm 38$ | 46.25 | $43.4 \pm 49.4$ |
| TR+W-Voting Chen & Li (2024) | 87.8 | $38.4 \pm 29.8$ | 72.1 | $46.9 \pm 38$ | 56.75 | $43.4 \pm 49.4$ |
| *Staple*: after offline optimization, performing reasoning with plan trees | | | | | | |
| Direct | 77.56 | $0.5 \pm 0.07$ | 53.11 | $1.2 \pm 0.09$ | 45.62 | $1.4 \pm 0.1$ |
| D-Contrastive-5 | 78.74 | $3.5 \pm 1.2$ | 58.11 | $4.7 \pm 1.6$ | 47 | $5.1 \pm 1.9$ |
| D-Contrastive-10 | 77.95 | $13.7 \pm 5.6$ | 59.22 | $8.7 \pm 2.8$ | 47.12 | $8.9 \pm 3.3$ |
| Ensemble-5 | 81.89 | $2.2 \pm 0.6$ | 58.56 | $4.5 \pm 1.9$ | 48.38 | $4.3 \pm 1.3$ |
| Ensemble-10 | 83.07 | $4.1 \pm 1.3$ | 62.56 | $8.9 \pm 3.6$ | 50.62 | $9.2 \pm 2.7$ |
| Ensemble-20 | 84.25 | $8.2 \pm 3.2$ | 65.11 | $17.6 \pm 9.3$ | 51.75 | $15.4 \pm 2.1$ |
| Ensemble-40 | 85.04 | $17.4 \pm 11.2$ | 66.44 | $35.2 \pm 12.5$ | 52.5 | $32.3 \pm 15.5$ |
| Ensemble-80 | 85.04 | $33.2 \pm 17.5$ | 66.67 | $70.63 \pm 28.5$ | 52.5 | $65.3 \pm 23.4$ |
| Adaptive | 84.25 | $4.6 \pm 1.9$ | 66.78 | $6.3 \pm 2.3$ | 52.5 | $6.6 \pm 2.4$ |

highest $r_{llm}$ and $r_w$ during optimization. Thirdly, as illustrated in Fig. 1, by incorporating more training samples in the optimization process, *Staple* constructs plan trees with more nodes/plans, and importantly, the solving rate increases significantly. This result indicates that *Staple* learns and evaluates various reasoning logics, thereby gaining reliable plans by leveraging GPT-4 to reason on more training samples rather than merely adding nodes. Ultimately, each retrieved plan is presented as human-readable text, rendering the reasoning process interpretable and reliable, as discussed in Section 4.2 and sections B D of the Appendix.

## 4.2 PLAN DATABASE: REUSABLE, UPDATABLE AND INTERPRETABLE

The three plan databases generated through *Staple* demonstrate the once-for-all property, indicating that subsequent universal tasks can utilize these plan trees directly without extra effort. First, Table 2 illustrates that we can reuse the plan tree $\mathbf{P}_A$ of category `Algebra` across three datasets. Retrieving plans from $\mathbf{P}_A$ optimized on $\mathtt{U}_{800}$ to solve 254 test questions from $\mathtt{U}_{254}$ achieves nearly the same solving rate as the $\mathtt{U}_{254}$'s own policy tree. Second, when GPT-3.5 and Llama 2 retrieve from $\mathbf{P}_A$ optimized with GPT-4 to answer `Algebra` questions, these older LLMs experience a substantial improvement in their solving rate, such as the 47.06 for GPT 3.5 on $\mathtt{T}_{51}$. However, a limitation of *Staple* emerges when applying policy trees across categories. This suggests that once the policy trees are optimized on $\mathtt{M}\text{-}\mathtt{G}_{500}$, $\mathtt{M}\text{-}\mathtt{NT}_{500}$, and $\mathtt{T}\text{-}\mathtt{K}_{29}$, other categories may not benefit from them as the solving rates are close to 0, as demonstrated in Table 2.

The continuously updatable feature in *Staple* allows it to accumulate plans that capture a domain's general knowledge by analyzing additional samples. Specifically, updating $\mathbf{P}_A$ using Algebra samples from $\mathtt{U}$ and $\mathtt{M}$ enhances the solving rates for this category across three datasets accordingly. Notably, for $\mathtt{M}$, which has only 51 available samples, directly reusing this updated policy tree results in a 94.12 solving rate.

Fig. 2 illustrates a detailed online plan searching process. The process begins with step **0**, where a tag-matched plan tree is retrieved from the `MATH` plan database. By comparing the question with Plan Thought $\Upsilon_0$, *Staple* calculates and selects $N$-24 $P$-1, which has the highest priority score of 1.57. Using this selected plan as a guide, GPT-4 generates a reliable reasoning step 1, represented as $N$-1 $S$-1 in the thought structure, where $S$-1 indicates Step 1. Next, Step 1 is compared with $\Upsilon_1$ to execute the **2. Best first search**, resulting in $N$-56

Table 2: Utilizing the plan tree in *Staple* optimized with GPT-4 on four cases. Here, the three datasets, including `AQUA-RAT`, `MATH`, `TheoremQA`, are abbreviated as U, M, and T to save space. The subscripts indicate the amount of data used for optimization and testing. And the category of plan tree $P_A$ used here is Algebra ($A$). $G$ $NT$ $K$ are the abbreviations of Geometry, Number Theory, and Kinetics, respectively.

| Purpose | Reuse $P_A$ across dataset | | | Reuse across category | | | Reuse $P_A$ across LLMs | | Update $P_A$ across dataset | |
|---|---|---|---|---|---|---|---|---|---|---|
| Dataset | U$_{800}$ | M$_{700}$ | T$_{51}$ | M-G$_{500}$ | M-NT$_{500}$ | T-K$_{29}$ | GPT3.5 | Llama2 | U+M | U+M+T |
| U$_{254}$ | 84.25 | 87.71 | 69.1 | 4.7 | 2 | 3.5 | 53.54 → 76.38 | 18.5 → 47.24 | 87.4 | 87.4 |
| M$_{500}$ | 84 | 83 | 70 | 2 | 1.6 | 2.2 | 51.2 → 76.4 | 20.2 → 34 | 86 | 86.6 |
| T$_{51}$ | 88.24 | 84.31 | 58.8 | 5.8 | 3.9 | 1.9 | 27.45 → 74.51 | 17.65 → 35.29 | 94.12 | 96.08 |

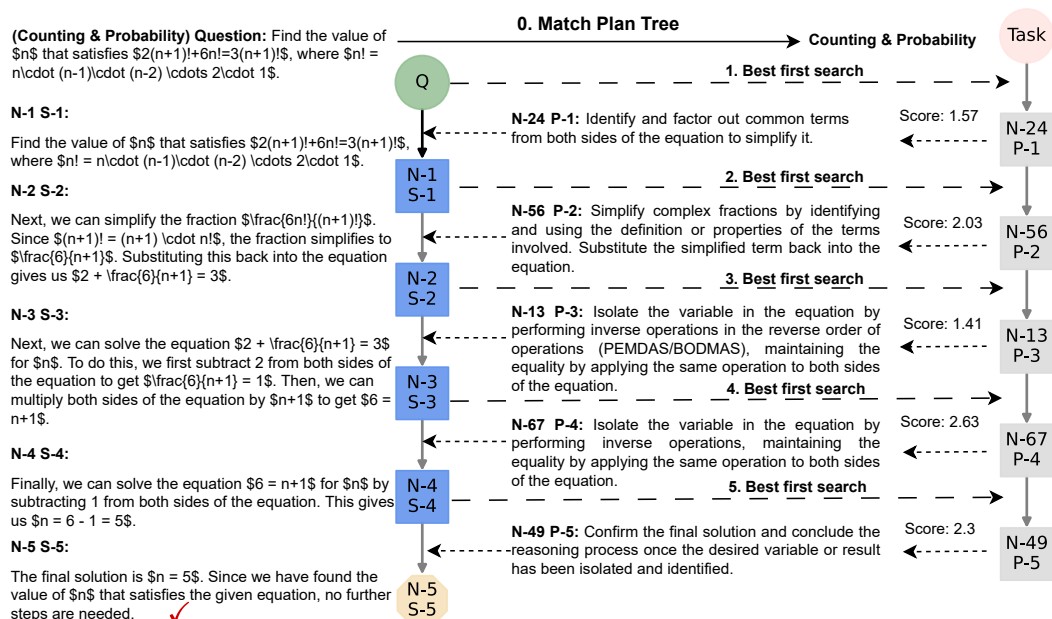

Figure 2: Illustrating how GPT-4 with *Staple*-Adaptive tackles a question from the category "Counting & Probability" of the `MATH` data during reasoning. Refer to Fig. 4 for explanations of these plotted modules.

$P$-2 as guidance for reasoning step 2. This adaptive searching mechanism is repeated 5 times, ultimately yielding the final answer $n = 5$. Unlike existing approaches such as TR Chen & Li (2024), STEPBACK Zheng et al. (2024b), and CR Zhang et al. (2023a), the reliable plans discovered by *Staple* during reasoning offer human-readable and insightful information about how LLMs conduct an effective reasoning process.

## 5 CONCLUDING REMARKS

In this paper, we proposed *Staple*, a new plan retrieval augmented reasoning framework designed to optimize general-purpose reasoning instructions and retrieve appropriate plans to guide large language models towards reliable problem-solving. The *offline plan optimization* stage in *Staple* leverages LLMs to autonomously explore and refine the plan space, structuring it as a tree based on Monte Carlo Tree Search. Subsequently, the *online plan searching* stage adaptively identifies the optimal plan for each reasoning step, ensuring dependable multi-step reasoning even in the presence of hallucinations. Upon optimizing the plan tree for each task category, *Staple* generates a reusable plan database that can be updated and applied to downstream problems. Our experiments demonstrate that *Staple* consistently achieves the second-highest solving rate while maintaining minimal resource consumption. Finally, the human-interpretable plans derived from tasks and employed in the reasoning process offer valuable insights into the logical processes of LLMs, potentially yielding broader implications for the field.

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

## A  REPRODUCIBILITY

Full reproducibility is essential for effective work, particularly when projects involve reliable reasoning with Large Language Models (LLMs). To facilitate this, this section provides detailed instructions on the Source Code (Section A.1), Result Reproduction (Section A.2), and Prompts (Section A.3) for readers to successfully reproduce the results of the paper. Specifically, in Section A.4, we offer an illustration showcasing the components that appear in the figures, highlighting thought structures, plan trees, and plan searching in the paper. Throughout the appendix, all plan tree-related figures are generated using Networkx Hagberg et al. (2008).

### A.1  SOURCE CODE

Our implementation of *Staple* is based on a new Python framework we developed, called *Llmpebase*. The source code, along with a comprehensive explanation of its module structure and functions, is available in the `code/` folder of the supplementary materials. We have designed the code for ease of use, allowing *Staple* to be run in less than one minute to facilitate quick testing.

In addition, we have implemented several representative competitors mentioned in the experiments. These include Zeroshot-COT, Chain Reasoning, ToT reasoning Yao et al. (2023), BoT reasoning Chen et al. (2024), and TR Chen & Li (2024), all of which can be found in the *code/* directory.

Our framework, *Staple*, is referred to as `StapleReasoning` in the code. The code includes all necessary modules, including our designs and visualization. Some of them are as follows:

- embedder.py: Compute the text embeddings and Search Top-K neighbors.
- mcts_thought_structure.py: Perform the core MCTS-based plan optimization in according to Algorithm 1.
- optimize_pipeline.py: Implement the *offline plan optimization*.
- plan_tree.py: Define the plan tree structure and operations.
- staple_prompts: Implement all prompts used in *Staple*.
- staple_system_prompts: Implement the system prompts used in *Staple*.
- StapleOptimization.py: Running interface.
- StapleRetrieval.py: Running interface for *online plan searching*.

### A.2  RESULT REPRODUCTION

*Staple* is capable of being executed by using a single line of command. For example, to perform *offline plan optimization* on the `AQUA-ART` dataset, one can run the following command:

```
python examples/StapleReasoning/StapleOptimization.py -c
configs/AQUA/GPT4/StapleReasoning_ZeroshotCoT.yml -b ICLR
```

while the command for `MATH` is:

```
python examples/StapleReasoning/StapleOptimization.py -c
configs/MATH/GPT4/StapleReasoning_ZeroshotCoT.yml -b ICLR
```

Throughout the optimization process, our code automatically generates and stores all intermediate results. These include the thought structure, plan trees, reasoning process, solution, token/interaction cost, and the final plan database. When executing the aforementioned commands, the results are saved in the `ICLR/results/` and `ICLR/results/visualizations` folders. Within these directories, results can be located in the corresponding subfolder named after the reasoning method, such as `StapleReasoning__*`. For each question's optimization, the file is saved as `thought_structure-Epoch 1-Idx 4-ID<5026>`. Additionally, the solution and token/interaction cost are stored in the `llm_records` file. Notably, the plan database is stored in the `PlanBase` folder. Detailed information can be found in Fig. 3.

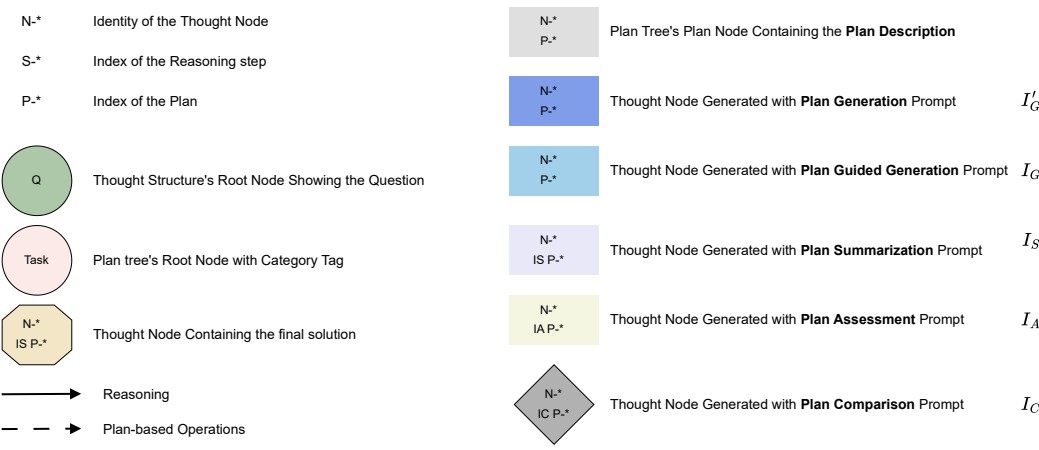

Figure 3: Illustrating the positions and details of the results obtained by *Staple*.

| | | |
|---|---|---|
| N-* | Identity of the Thought Node | |
| S-* | Index of the Reasoning step | |
| P-* | Index of the Plan | |

Q — Thought Structure's Root Node Showing the Question

Task — Plan tree's Root Node with Category Tag

N-* IS P-* — Thought Node Containing the final solution

→ Reasoning

--→ Plan-based Operations

N-* P-* — Plan Tree's Plan Node Containing the **Plan Description**

N-* P-* — Thought Node Generated with **Plan Generation** Prompt $\mathbb{I}'_G$

N-* P-* — Thought Node Generated with **Plan Guided Generation** Prompt $\mathbb{I}_G$

N-* IS P-* — Thought Node Generated with **Plan Summarization** Prompt $\mathbb{I}_S$

N-* IA P-* — Thought Node Generated with **Plan Assessment** Prompt $\mathbb{I}_A$

N-* IC P-* — Thought Node Generated with **Plan Comparison** Prompt $\mathbb{I}_C$

Figure 4: Illustrating the modules used during plotting the reasoning process of *Staple*.

## A.3 PROMPTS

All prompts utilized by *Staple* are located in `code/StapleReasoning/staple_prompts.py` and `code/StapleReasoning/staple_system_prompts.py`. We present the detailed prompts here to provide further clarification.

**Prompt for the Plan Guided Thought Generation** $\mathbb{I}_G$:

- System Prompt. As an expert in problem-solving, you are skilled in methodical, step-by-step reasoning guided by policies, each presenting a general-purpose reasoning instruction for one step. The plan is a high-level, question-agnostic principle that facilitates deducing a single logical reasoning step toward addressing one task. Following the plan, you should generate a specific reasoning step. Start by reviewing the problem, the previous reasoning steps, and their corresponding policies, then follow the given specific plan to directly generate the next step. Remember, your next step should include a precise analysis and the corresponding mathematical expression. This comprehensive approach will ensure a thorough solution. Utilize Python Programming as an auxiliary tool when necessary.

- {Question}$\backslash n$Let's focus on following the plan to directly generate the next reasoning step for the reasoning steps below.$\backslash n\backslash n\{\boldsymbol{z}_n\}\backslash n\{\boldsymbol{\psi}_n\}\backslash n\backslash n\{\psi_{(n+1)*}\}\backslash n\backslash n$Please review the reasoning steps along with their plans, then follow the Plan $\psi_{(n+1)*}$ to proceed to directly generate the best next step, i.e., Step {n+1}.

**Prompt for the Plan Exclusion Generation** $\mathbb{I}_E$:

- System Prompt. As an expert in problem-solving, you are adept at methodical, step-by-step reasoning while avoiding duplicating the given policies, each presenting a general-purpose reasoning instruction for one step. You need to know that each plan is a high-level, question-agnostic principle that facilitates deducing a single logical reasoning step toward addressing one task. Thus, excluding the plan means

having a new and different plan to generate the corresponding next step. Remember, your response should only include one next step. Start by reviewing the problem and reasoning steps, then exclude the given specific policies to generate the next step. You can ignore the plan exclusion when no plan is given. The next step should contain the precise analysis and the corresponding mathematical expression. Utilize Python Programming as an auxiliary tool when necessary.

- $\mathbb{I}_E$: {Question}$\backslash n$Let's focus on avoiding using the given policies to carefully and directly generate the next possible reasoning step for the reasoning steps below.$\backslash n\backslash n\{\boldsymbol{z}_n\}\backslash n\{\boldsymbol{\psi}_n\}\backslash n\backslash n\{C\left(\psi_{n*}\right)\}\backslash n\backslash n$Please review the reasoning steps and their policies, then specifically avoid repeating all Plans $C\left(\psi_{n*}\right)$ to proceed to directly generate the best next step, i.e., Step {n+1}.

**Prompt for the Plan Comparison $\mathbb{I}_C$:**

- System Prompt. As a professional plan comparison expert, your expertise lies in judging whether a plan exists in a plan pool containing various policies. Remember that plan is a general-purpose reasoning instruction and is a high-level, question-agnostic principle. Please perform the comparison in terms of the logic, high-level ideas, theorems, or rules. Please compare the given plan with each of the policies in the pool. Once there is a similar one, return True. Start by reviewing policies in the pool, then directly judge whether the given plan already exists. The output should be either True or False.

- $\mathbb{I}_E$: Let's focus on whether the given plan exists in the plan pool.$\backslash n\backslash n\{\psi_{n+1}\}\backslash n\backslash n\backslash n\{C\left(\psi_{n*}\right)\}\backslash n\backslash n$Please judge whether the Plan $\psi_{n+1}$ already exists in the $C\left(\psi_{n*}\right)$. Only output True if exists, or False if not. Remember that plan is a high-level, question-agnostic principle. Do not focus on text details but on the logic, high-level ideas, theorems, or rules.

**Prompt for the Plan Assessment $\mathbb{I}_A$:**

- System Prompt. You are a professional mathematician with expertise in assessing a plan that presents general-purpose reasoning instruction for generating the next reasoning step. Specifically, the plan is a high-level, question-agnostic principle that facilitates deducing a single logical reasoning step toward addressing one task. You should assess the plan by scoring it based on whether it guides generating the reasonable reasoning step that progresses the problem-solving. Start by reviewing the given problem, reasoning steps already taken, and the generated next step guided by the plan, then directly assess this given plan. Importantly, the generated reasoning step guided by this plan is also given to facilitate the assessment. Utilize Python Programming as an auxiliary tool when necessary. The output should be a float score without including any other content.

- $\mathbb{I}_E$: {Question}$\backslash n$For the given question, Let's focus on assessing whether the plan can guide the generation of an effective next reasoning step.$\backslash n\backslash n\{\boldsymbol{z}_n\}\backslash n\{z_{n+1}\}\backslash n\backslash n\{\psi_{n+1}\}\backslash n\backslash n$Please review the reasoning steps already taken and the generated next Step $\{z_{n+1}\}$ guided by the Plan $\{\psi_{n+1}\}$, then assess this Plan $\{\psi_{n+1}\}$.

**Prompt for the Plan Summarization $\mathbb{I}_S$:**

- System Prompt. You are an expert in identifying, extracting, and summarizing the plan that underpins one reasoning step. The summarized plan should be a general-purpose reasoning instruction and, thus, is a high-level, question-agnostic principle. Please get such a plan containing the highest-level ideas, principles, rules, or theorems from the given reasoning step. Start by reviewing the given question and any previous reasoning steps already taken along with their corresponding policies, then directly summarize the plan of the given reasoning step. Please summarize the plan directly and briefly, avoiding including the specific contents of the given question or any reasoning steps.

- $\mathbb{I}_S$: {Question}\nFor the given question, let's focus on summarize the plan that underpins the reasoning step {}.\n\n{$z_n$}\n{$\psi_n$}\n\n{$z_{n+1}$}\n\nPlease review the reasoning steps and their corresponding policies and proceed to summarize the plan of Step $z_{n+1}$, i.e., Plan {n+1}.

- One Example from `MATH` Dataset:

```
Input: \nQuestion: Find the number of ordered pairs of positive
    integers $(a,b)$ such that $a+b=1000$ and neither $a$ nor $b$ has
     a zero digit.\n \n\nFor the given question, let's focus on
    summarize the plan that underpins the reasoning step 2.\n\n<Step
    Chain>\n Step 1. Recognize that the problem is a partition
    problem, where we need to partition the number 1000 into two
    parts, each represented by a positive integer. However, the
    condition that neither of the integers can contain a zero digit
    adds a layer of complexity. To tackle this, we can start by
    finding the total number of ways to partition 1000 into two
    positive integers without any restrictions. This can be done by
    subtracting 1 from 1000, as the number of ways to partition a
    number n into two parts is n-1. This gives us 999 ways.\t\n<\\
    Step Chain>\n<Plan Chain>\n Plan 1. Identify the problem as a
    partition problem and calculate the total number of ways to
    partition the given number into two parts without any
    restrictions. This is typically done by subtracting 1 from the
    given number.\n<\\Plan Chain>\n\n<Step>\n Step 2. The next step
    is to determine the number of ways in which a zero digit can
    appear in either $a$ or $b$. This can occur if $a$ or $b$ is a
    multiple of 10. We need to subtract these cases from the total
    999 ways. The multiples of 10 between 1 and 1000 are 10, 20, 30,
    ..., 990, 1000. There are 100 such numbers. However, we need to
    exclude the case where $a$ or $b$ is 1000, as it is not a valid
    partition of 1000 into two positive integers. Therefore, there
    are 99 ways in which a zero digit can appear in either $a$ or $b$
    .\n<\\Step>\n\nPlease review the reasoning steps within <Step
    Chain> and their corresponding policies within <Plan Chain> and
    proceed to summarize the plan of Step 2 within <Step Chain>, i.e
    ., Plan 2. Only direct output summarized plan. Do not include the
     Plan index in the output. Remember that the plan is a high-level
    , question-agnostic principle. Do not include any question or
    reasoning step content in the plan.
Output: Identify the cases that violate the given conditions and
    calculate their number. Subtract these cases from the total
    number of possibilities to get the number of valid cases.
```

### A.4 PLOT ILLUSTRATION

As previously stated, our code generates plots for all intermediate results, encompassing thought structures and plan trees. Furthermore, we illustrate the MCTS-based optimization process in *Staple* as a component of the thought structure to clearly demonstrate its functionality, as elaborated in Section B. Consequently, Fig. 4 shows the significance of each module within the figures.

## B EXAMPLES OF OPTIMIZATION

Based on the experimental results in the supplementary material folder `MATH-explain-1/`, we can provide a detailed explanation of how *Staple* utilized GPT-4 to optimize the plan tree for the task "Counting & Probability". In the initial stage of offline optimization, the 3rd iteration of Algorithm 1 explores, optimizes, and creates 6 new plans for the Plan Tree $\Psi$, which contains 2 paths and 13 plans. The value function described in Section 3.2 is defined for arbitrary thoughts. Specifically, based on $r_{llm}$ and $r_w$ defined in Reward assignment and $K(\cdot)$ defined in Reward assignment, we present that the value function includes three parts:

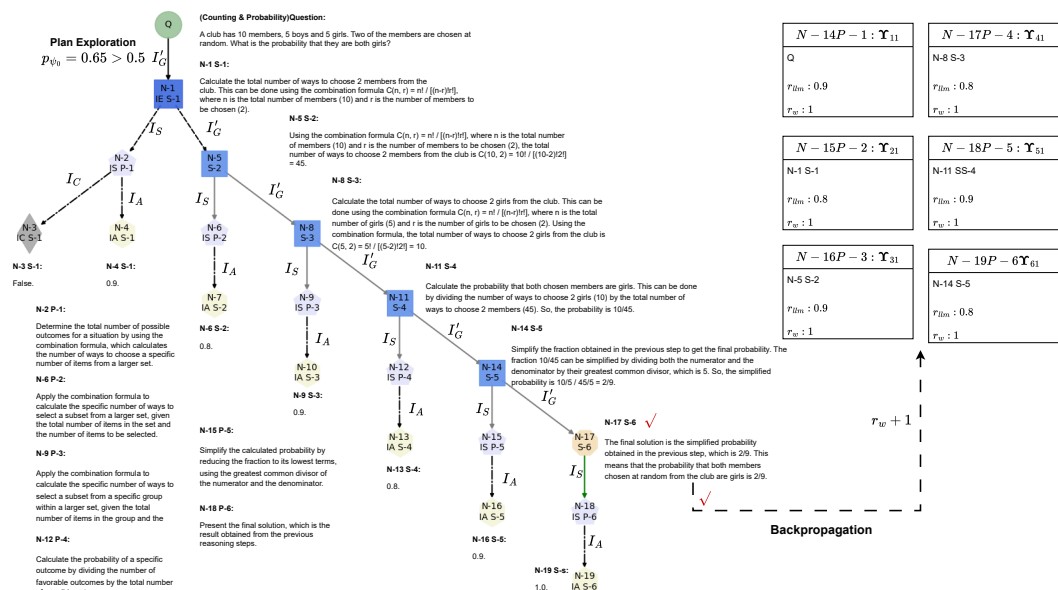

Figure 5: Illustration of the 3rd iteration of *offline plan optimization* in *Staple*. This figure shows the exact process of the Algorithm 1 by presenting how the reasoning is performed and how the plans are explored, summarized, assessed, and created. The table in the figure shows the specific contents of the Plan Thought $\Upsilon$ in each node of the plan tree and how it is updated during optimization. These tables are extracted from the Plan Tree $\Psi$ shown in Fig. 6. We present them here to make a clear alignment with the optimization process.

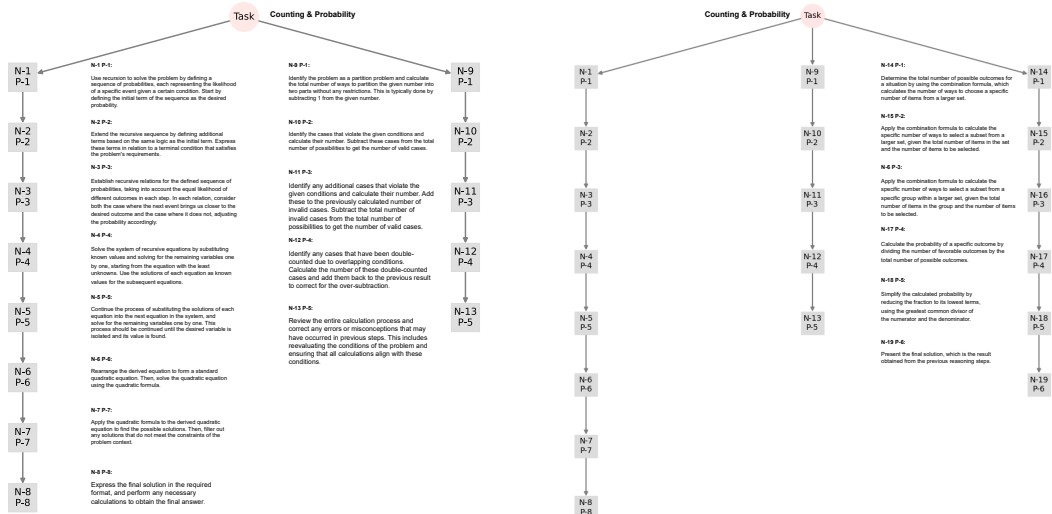

Figure 6: Illustration the specific plans in Plan Tree $\Psi$ and the tree change before and after the third optimization iteration.

- $r_{llm}$ represents the evaluation score of the next thought generated by the LLM based on the current thought.

- $r_w$ indicates whether selecting the plan for the next thought generation from the current thought results in the final correct solution. It is used for backpropagation in MCTS.

- $K(\cdot)$ represents the similarity between the current thought and all previous thoughts that led to the corresponding plan.

With this value function, we aim to provide an evaluation score for selecting a plan for the current thought, based on both past experience and the LLM's self-assessment.

As shown in Fig. 5, the Plan Tree $\boldsymbol{\Psi}$ tagged with "Counting & Probability" is retrieved based on the category of the question $Q$. The Plan Exploration is then performed by computing the exploration probability $p_{\psi_0} = 1/\left(1 + e^{0.2(2-5)}\right) = 0.65$, where 2 represents the number of child plans of $\psi_0$ as shown in the 1st depth of the policy tree in Fig. 6. Using the prompt $\mathbb{I}'_G$, GPT-4 explores new plans by excluding existing ones. For the generated thought $N-1S-1$, GPT-4 with $\mathbb{I}_S$ summarizes the plan, which is then assessed using the prompt $\mathbb{I}_A$ to generate $r_{llm}$. Subsequently, GPT-4 with $\mathbb{I}_C$ compares the summarized plan with existing plans at the 1-st depth, namely $N-1P-1$ and $N-9P-1$ of Fig. 6. As this is a new plan, a new node $N-14P-1$ is created in $\boldsymbol{\Psi}$ to record the plan's content and the Plan Thought $\boldsymbol{\Upsilon}_{11}$. For instance, in Fig. 5, the table $N-14P-1 : \boldsymbol{\Upsilon}_{11}$ records the previous thought $Q$ and its corresponding $r_{llm}$. Following this expansion phase in MCTS, we conduct the simulation/rollout by allowing GPT-4 to continue reasoning until it reaches a solution $N-17S-6$. During this process, a sequence of plans is summarized and assessed, resulting in the creation of nodes such as $N-15P-2$, $N-16P-3$, $N-17P-4$, $N-18P-5$, and $N-19P-6$, each with their respective $r_{llm}$. Consequently, $\boldsymbol{\Psi}$ develops a new plan path where each node/plan has its own Plan Thought $\boldsymbol{\Upsilon}$, as illustrated in Fig. 5 and Fig. 6. After comparing the obtained solution 2/9 with the ground truth $\frac{2}{9}$, *Staple* backpropagates the $r_w : 1$ to each node of the plan path and updates the Plan Tree $\boldsymbol{\Psi}$ and Plan Thought $\boldsymbol{\Upsilon}$ accordingly.

Fig. 6 illustrates the specific content of a $\boldsymbol{\Psi}$, which comprises numerous plan combinations presented as the paths in the tree. The left subfigure displays the policy tree with 2 paths and 13 nodes prior to the third iteration of optimization. It is evident that each plan serves as a concise and high-level principle outlining *what to do* during the reasoning step. By using the plan as an instruction in the prompt, LLMs are guided to follow a specific logical sequence to reason step-by-step toward the solution. Significantly, after incorporating the new plan path into the tree, plans denoted as $P-1$ at the first depth of the tree differ from one another, indicating that each plan represents a unique approach to addressing the given question. Consequently, by continuously conducting trial reasoning with LLMs and optimizing the policy tree, we can explore a task's plan space to acquire effective plans with priority scores that can be directly utilized by downstream tasks.

## C  LIMITATIONS

As an inherent limitation, *Staple* still incurs token and time costs during the offline plan optimization stage. Specifically, we use the training sets from the `AQUA-RAT`, `MATH`, and `TheoremQA` datasets, with sample sizes of 800, 700, and 800, respectively. As outlined in the experimental settings, we perform 10 epochs to construct the plan trees. Consequently, in the offline stage, the total token costs are 5.811M, 7.441M, and 8.217M, where M represents 1 million tokens, with corresponding time costs of 36 minutes, 58 minutes, and 1 hour and 23 minutes. The time costs for `MATH` and `TheoremQA` are relatively high due to the complexity of the questions, as LLMs generally need to perform multiple reasoning steps to solve them.

*Staple*'s applicability covers two scenarios: Case A, where users want to create the plan database from scratch, and Case B, where users first download a pre-learned plan database from Hugging Face and then apply it to their own task.

In summary, for users with weak LLMs who want to create a plan database from scratch, *Staple* offers only limited improvement, as the database generated by weak LLMs may not effectively guide problem-solving. However, for users with weak LLMs who reuse a plan database generated by top-performing LLMs, such as GPT-4, *Staple* can significantly enhance performance by retrieving effective plans from the downloaded database. The experimental results presented below support our argument. Specifically, we use the Number Theory subset of the MATH dataset, consisting of 800 samples for plan optimization and 540 samples for evaluation. The offline plan optimization of *Staple* is conducted using the 800 training set samples.

Table 3: Comparing the performance of *Staple* with different weak LLMs in Case A and Case B. The 4-shot CoT Wei et al. (2022) is the baseline. The "*Staple* from scratch" column means that the model is used in both the offline plan optimization and online plan searching. The "Reuse the plan database" column means that during the online plan searching, the model retrieves plans from the plan database generated by *Staple* using GPT-4.

| Models | 4-shot CoT | *Staple* from scratch | Reuse the plan database |
|---|---|---|---|
| Llama2 7B | 12.59 | 12.04 | 29.81 |
| Llama2 13B | 26.67 | 29.26 | 40 |
| Llama2 70B | 34.26 | 38.7 | 49.63 |
| Llama3 8B | 28.89 | 29.81 | 44.44 |
| Llama3 70B | 48.7 | 53.15 | 63.7 |

Thus, the results shown in Table 3 lead to the following three conclusions.

- For Case A, with weak LLMs, *Staple* offers only limited improvement. This may restrict its usability when users prefer to execute both the offline and online stages of *Staple* using weak LLMs.

- For Case A, as the capability of the LLMs increases, the performance of *Staple* improves accordingly.

- For Case B, users with weak LLMs can significantly enhance their performance by reusing the plan database from top-performing LLMs, such as GPT-4. The trees in the plan database of our *Staple* are in text format, making the database easy to share on Hugging Face. This further highlights the practicality of our *Staple*, enabling researchers worldwide to perform reliable multi-step reasoning by downloading the well-optimized plan database at no additional cost.

## D   Plan Tree of the Intermediate Algebra of `MATH` dataset

Based on the folder `MATH-plantree-IntermediateAlgebra/` in the supplementary materials, we illustrate the plan tree for the Intermediate Algebra task in the `MATH` dataset. The plan tree is optimized by *Staple* following Algorithm 1 with the parameters mentioned in Sec. 4.

### D.1   Optimized Plans

**Plans sampled from the 1-depth**:

- $N-1P-1$: Recognize the properties of a polynomial, specifically that the sum of the roots is equal to the negation of the coefficient of the second highest degree term divided by the coefficient of the highest degree term. Use this property to set up an equation and solve for the desired variable.

- $N-15P-1$: Identify the given properties of the geometric figures and use relevant formulas to form equations. Use these equations to solve for unknown variables.

- $N-30P-1$: Rewrite the given equation in a standard form by grouping related terms together and completing the square. This involves identifying and grouping similar terms, then adjusting the equation to form perfect squares.

- $N-36P-1$: Identify and categorize the types of given equations in a problem. Recognize that the intersection points of the graphs represented by these equations will be the solutions to the system formed by these equations.

- $N-36P-1$: Identify the given inequality and the conditions provided. Apply the mathematical principle that when a negative number is multiplied on both sides of an inequality, the direction of the inequality reverses. Compare the resulting inequality with the given inequality to determine if the statement is always true.

- $N-76P-1$: Identify the roots of the polynomial from the given conditions. Express the polynomial in its factored form using these roots.

- $N - 82P - 1$: Recognize that if a polynomial has real coefficients, its non-real roots must come in conjugate pairs. If a complex number is a root, then its conjugate must also be a root.
- $N - 91P - 1$: Start the process of polynomial long division by dividing the first term of the numerator by the first term of the denominator to obtain the first term of the quotient.

**Plans sampled from the 2-depth**:

- $N - 2P - 2$: Simplify the equation by combining like terms to make it easier to solve for the desired variable.
- $N - 16P - 2$: Express one variable in terms of another using one equation, then substitute this expression into another equation to reduce the number of unknowns and simplify the problem.
- $N - 25P - 2$: Substitute the identified parameters into the appropriate formula or equation to calculate the desired value or result.
- $N - 31P - 2$: Calculate the values needed to complete the square for each variable in the equation. This involves taking half of the coefficient of each variable, squaring it, and adding it to both sides of the equation. This will transform the equation into a form where the variables are part of perfect squares.
- $N - 45P - 2$: Identify the conditions under which the rearranged inequality holds true by leveraging known mathematical principles or properties. Solve the resulting inequality to find the range of possible values for the variables or constants involved.
- $N - 37P - 2$: Substitute one equation into another when trying to find the intersection points of two graphs. This simplifies the system of equations into a single equation with one variable, which can then be solved.
- $N - 50P - 2$: Apply the mathematical principle that when a negative number is multiplied on both sides of an inequality, the direction of the inequality reverses. Compare the resulting inequality with the given inequality to verify if the statement is always true.
- $N - 56P - 2$: Distribute the terms in the equation to simplify it further.
- $N - 64P - 2$: Set each factor of the factored equation equal to zero and solve for the variable to find the solutions of the equation.
- $N - 70P - 2$: Identify and list all the factors of the given numbers, considering both positive and negative values.

**Plans sampled from the 3-depth**:

- $N - 3P - 3$: Separate the real and imaginary parts of a complex equation to form two separate equations. Use these equations to solve for the desired variables. If an inconsistency or error is found, consider revisiting previous steps to identify and correct the mistake.
- $N - 17P - 3$: Substitute the expression of one variable in terms of another into the equation, then simplify the equation to solve for the desired variable.
- $N - 26P - 3$: Perform arithmetic operations to simplify the expression or equation, if necessary, to get the final result.
- $N - 32P - 3$: Normalize the equation to the standard form of an ellipse by dividing all terms by the constant on the right side of the equation. This will allow for the identification of the square of the semi-major and semi-minor axes.
- $N - 36P - 3$: Identify and categorize the types of given equations in a problem. Recognize that the intersection points of the graphs represented by these equations will be the solutions to the system formed by these equations.
- $N - 46P - 3$: Solve the derived inequality by isolating the variable or constant of interest. Use appropriate mathematical operations to manipulate the inequality, considering both positive and negative solutions if necessary.

- $N - 51P - 3$: Apply the mathematical principle that when a negative number is multiplied on both sides of an inequality, the direction of the inequality reverses. Compare the resulting inequality with the given inequality to ascertain if the statement is always true.

- $N - 57P - 3$: Combine like terms in the equation to further simplify it.

- $N - 65P - 3$: Identify patterns or properties in the simplified expression that allow for further simplification, such as telescoping series where most terms cancel out, leaving only a few terms to compute.

**Plans sampled from the $4$-depth**:

- $N - 4P - 4$: Identify and correct any errors in previous steps, particularly in mathematical calculations or the application of formulas. If an inconsistency is found, revisit the steps to ensure the correct separation of real and imaginary parts in complex equations. Use the corrected equations to solve for the desired variables.

- $N - 18P - 4$: Simplify the equation by cancelling out common factors or terms, in order to isolate and solve for the desired variable.

- $N - 27P - 4$: Continue to perform arithmetic operations to further simplify the expression or equation, if necessary, to get the final result.

- $N - 33P - 4$: Identify the formula for the area of an ellipse and apply it by substituting the lengths of the semi-major and semi-minor axes obtained from the standard form of the ellipse equation. This involves taking the square root of the values of a$\hat{2}$ and b$\hat{2}$ to get the lengths of the axes, and then substituting these values into the area formula.

- $N - 47P - 4$: Identify the maximum or minimum value from the range of possible values for the variable or constant of interest, based on the requirements of the problem.

- $N - 39P - 4$: Further simplify the equation by combining like terms involving the same variables on both sides. This process often results in a more manageable equation, such as a quadratic equation, which can then be solved.

- $N - 52P - 4$: Apply the mathematical principle that adding the same number to both sides of an inequality does not change the direction of the inequality. Compare the resulting inequality with the given inequality to determine if the statement is always true.

- $N - 58P - 4$: Isolate the terms involving the variable on one side of the equation and the constant term on the other side by using the principle of equality to add or subtract the same quantity from both sides of the equation.

- $N - 72P - 4$: Simplify the fractions obtained from the previous step by dividing each numerator by each denominator and reducing the resulting fraction to its simplest form. Count the total number of unique simplified fractions to determine the number of different possible outcomes based on the given conditions.

- $N - 139P - 4$: Summarize the results of the verification process and conclude the final solution based on the validated roots.

**Plans sampled from the $6$-depth**:

- $N - 6P - 6$: Substitute the expression for a variable obtained from previous steps into the given equations or expressions to further simplify or solve them.

- $N - 20P - 6$: Use the formula for the perimeter of a geometric figure, substituting in the known values of the variables, to calculate the perimeter.

- $N - 29P - 6$: Present the final solution or result obtained from the previous steps.

- $N - 35P - 6$: Simplify the final expression to obtain the final solution.

- $N - 41P - 6$: Apply the quadratic formula to solve for the variable in a quadratic equation. This involves identifying the coefficients of the quadratic equation and substituting them into the quadratic formula.

- $N - 54P - 6$: After evaluating each option individually according to the given conditions and mathematical principles, consolidate the results to identify which options are always true.

- $N - 60P - 6$: Set each factor of the factored equation equal to zero and solve for the variable to find the solutions of the equation.

- $N - 68P - 6$: Present the final numerical solution, which is the result of the previous calculations and simplifications.

**Plans sampled from the 9-depth**:

- $N - 9P - 9$: When faced with complex equations that are difficult to simplify directly, consider using the given conditions or properties of the problem to form new equations. Substitute these conditions back into the original equations or expressions and equate them to a known value, in this case zero. This process will yield a system of equations that can be solved simultaneously to find the desired variables.

- $N - 23P - 9$: State the final solution to the problem.

- $N - 44P - 9$: Interpret the mathematical results obtained from previous steps. If the results indicate an impossibility within the given mathematical system (such as the square root of a negative number in the real number system), conclude that there are no valid solutions. Use this conclusion to answer the original question or problem.

- $N - 90P - 9$: Present the final solution as the answer to the problem, indicating the conclusion of the reasoning process.

**Plans sampled from the 12-depth**:

- $N - 102P - 12$: Simplify the expression obtained from the previous step by combining like terms.

- $N - 136P - 12$: Conclude the reasoning process by stating the final solution, once all unknown variables have been determined.

- $N - 198P - 12$: Present the final solution by stating the quotient and the remainder obtained from the polynomial long division process.

- $N - 246P - 12$: After substituting the expressions for the variables into the equation, simplify the equation by cancelling out common terms. Continue to simplify the equation by expanding the terms and combining like terms.

- $N - 268P - 12$: Present the final solution of the problem.

## D.2 SOLVED QUESTIONS

In Fig. 7 and Fig. 8, we show how *Staple* guides GPT-4 to perform reliable step-by-step reasoning based on the searched step-wise plan. The initial experimental results are stored under folders `MATH-Questio-1/` and `MATH-Questio-2/` of the supplementary.

## E PLAN TREE OF THE KINETICS FOR THEOREMQA DATASET

### E.1 OPTIMIZED PLANS

**Plans sampled from the 1-depth**:

- $N - 1P - 1$ Identify the relevant formula or principle to solve the problem, in this case, the formula for escape speed. Substitute the given values into the formula to calculate the desired quantity.

- $N - 11P - 1$ Identify the problem as one of equilibrium and apply the principle of moments (torque) to set up an equation. The equation should reflect that the sum of the moments about any point is zero for the system to be in equilibrium. This

**Question:** Four horizontal lines and four vertical lines are drawn in a plane. In how many ways can four lines be chosen such that a rectangular region is enclosed?

*P-1: Identify the components needed to form the desired structure and determine the number of ways to select these components from the available set. Use the combination formula to calculate the number of ways to choose a specific number of items from a larger set.$*

**S-1:** The problem can be seen as a permutation problem involving movements in a grid with limited directions. In this case, we have four horizontal lines and four vertical lines drawn in a plane. To enclose a rectangular region, we need to choose two horizontal lines and two vertical lines. The total number of movements is four (two horizontal and two vertical), and there are two types of movements (horizontal and vertical). We can use the permutation formula for multiset to calculate the number of ways to arrange these movements. The permutation formula for a multiset is $n!/(n1! * n2! * ... * nk!)$, where n is the total number of items, and n1, n2, ..., nk are the numbers of each type of item. In this case, n = 4 (the total number of lines to be chosen), n1 = 2 (the number of horizontal lines to be chosen), and n2 = 2 (the number of vertical lines to be chosen). Therefore, the number of ways to choose four lines such that a rectangular region is enclosed can be calculated as $4!/(2! * 2!)$.

*P-2: Calculate the number of ways to select a subset from a larger set using the combination formula. This involves identifying the total number of items and the number of items to choose, and applying these values to the combination formula.*

**S-2:** Calculate the number of ways to choose two lines out of four for both horizontal and vertical lines separately. This can be done using the combination formula $C(n, r) = n! / [(n-r)! * r!]$, where n is the total number of items and r is the number of items to choose. In this case, n = 4 (the total number of lines) and r = 2 (the number of lines to choose). Therefore, the number of ways to choose two lines out of four can be calculated as $C(4, 2)$ for both horizontal and vertical lines.

*P-3: Apply the combination formula to calculate the number of ways to select a subset from a larger set for each category separately. Multiply the results together if selections from different categories are needed simultaneously.*

**S-3:** Calculate the number of ways to choose two lines out of four for both horizontal and vertical lines separately using the combination formula $C(4, 2) = 4! / [(4-2)! * 2!]$. This gives us 6 ways for each, horizontal and vertical lines. Now, since we need to choose 2 lines each from both horizontal and vertical lines, we multiply the two results together. Therefore, the total number of ways to choose four lines such that a rectangular region is enclosed is $6 * 6 = 36$.

*P-4: Confirm the correctness and completeness of the reasoning steps and solutions. If the final solution has been obtained and no further steps are needed, conclude the reasoning process.*

**S-4:** The reasoning steps provided are correct and complete. Since the final solution has been obtained in Step 3, no further steps are needed. The final solution is 36.

Figure 7: Illustration of how GPT-4 addresses the question guided by the plans retrieved from the plan tree of Intermediate Algebra.

involves identifying the forces acting on the system and their distances from the point of interest.

- $N - 17P - 1$ Identify the concepts relevant to the problem, in this case, static and kinetic friction. Understand that static friction must be overcome to initiate motion and kinetic friction must be overcome to maintain constant velocity. Use the appropriate equations for static and kinetic friction, recognizing that the normal force in these equations is equivalent to the weight of the object in question.

- $N - 23P - 1$ Identify the problem as one involving the conservation of energy, where potential energy is converted into kinetic energy. Apply the principle of conservation of energy, which states that the total energy in a closed system remains constant. Equate the initial potential energy to the final kinetic energy to establish a relationship for further calculations.

- $N - 31P - 1$ Identify the key concepts and relationships involved in the problem. Use the given information and these relationships to set up the necessary calculations.

- $N - 42P - 1$ Identify the variables in the problem and establish relationships between them. In this case, determine the time of flight for a moving object by expressing the distance it needs to travel in terms of its speed and the time of flight. This involves setting up an equation that equates the distance traveled to the product of the object's speed and the time of flight.

- $N - 50P - 1$ Apply the principle of conservation of momentum to establish an equation representing the relationship between the masses and velocities of the objects involved in the event.

**Plans sampled from the $2$-depth**:

- $N - 2P - 2$ Apply the identified formula or principle by substituting the given or known values into it to perform the necessary calculations.

- $N - 8P - 2$ Simplify the equation derived from the principle of equilibrium by eliminating common factors or variables, thus making the equation easier to solve.

**Question:** Find the real root of the equation $\sqrt{x} + \sqrt{x+2} = 10.$

*P-1: Apply the mathematical operation of squaring to both sides of an equation to eliminate square roots.*

**S-1:** To begin, we can square both sides of the equation to eliminate the square roots. This gives us:\n\n$[(\sqrt{x} + \sqrt{x+2})^2 = 10^2.]$.

*P-2: Apply the formula for the square of a binomial to expand the squared terms in an equation.*

S-2: We expand the left side of the equation using the formula for the square of a binomial $(a+b)^2 = a^2 + 2ab + b^2$. This gives us:\n\n$[x + 2\sqrt{x(x+2)} + (x+2) = 100.]$

*P-3: Isolate the term containing the square root on one side of the equation to simplify the equation further.*

S-3: To simplify the equation further, let's isolate the term with the square root on one side of the equation. This gives us:\n\n$[2\sqrt{x(x+2)} = 100 - 2x - 2.]$

*P-4: Simplify the equation by performing basic arithmetic operations to make it easier to solve.*

S-4: Simplify the right side of the equation to make the equation easier to solve. This gives us:\n\n$[2\sqrt{x(x+2)} = 98 - 2x.]$

*P-5: Divide both sides of the equation by a common factor to further isolate the term of interest.*

S-5: To isolate the square root term, divide both sides of the equation by 2. This gives us:\n\n$[\sqrt{x(x+2)} = 49 - x.]$

*P-6: Apply the mathematical operation of squaring to both sides of an equation to eliminate square roots.*

S-6: To eliminate the square root, square both sides of the equation again. This gives us:\n\n$[x(x+2) = (49 - x)^2.]$

*P-7: Apply the formula for the square of a binomial to expand the squared terms in an equation.*

S-7: Expand the right side of the equation using the formula for the square of a binomial $(a-b)^2 = a^2 - 2ab + b^2$. This gives us:\n\n$[x(x+2) = 49^2 - 2*49*x + x^2.]$

*P-8: Correct any mathematical errors in the previous steps to ensure the accuracy of the calculations.*

S-8: The right side of the equation was incorrectly expanded in the previous step. The correct expansion of $(49 - x)^2$ should be $2401 - 2*49*x + x^2$. So, we correct this to get:\n\n$[x(x+2) = 2401 - 2*49*x + x^2.]$

*P-9: Distribute and simplify terms in the equation to further simplify it.*

S-9: To simplify the equation further, let's distribute the x on the left side of the equation and simplify the right side of the equation. This gives us:\n\n$[x^2 + 2x = 2401 - 98x + x^2.]$

*P-10: Subtract the same term from both sides of the equation to simplify it further, especially when the term is present on both sides.*

S-10: The equation in the last step is a quadratic equation. However, we notice that the $x^2$ term is present on both sides of the equation. To simplify the equation further, we can subtract $x^2$ from both sides of the equation to eliminate the $x^2$ term. This gives us:\n\n$[2x = 2401 - 98x.]$

*P-11: Isolate the variable of interest by performing the same operation on both sides of the equation.*

S-11: To isolate the x term, add 98x to both sides of the equation. This gives us:\n\n$[100x = 2401.]$

*P-12: Divide both sides of the equation by the same number to solve for the variable of interest.*

S-12: To solve for x, divide both sides of the equation by 100. This gives us:\n\n$[x = \frac{2401}{100}.]$

*P-13: Simplify the expression or fraction to obtain the final value of the variable.*

S-13: The final step is to simplify the fraction on the right side of the equation to get the value of x. This gives us:\n\n$[x = 24.01.]$. The final solution is x = 24.01. This is the real root of the given equation.

Figure 8: Illustration of how GPT-4 addresses the question guided by the plans retrieved from the plan tree of Intermediate Algebra.

- $N - 28P - 2$ Apply the formula for static friction to calculate the coefficient of static friction. Use the given values for the static frictional force and the normal force in the calculation.

- $N - 44P - 2$ Formulate an equation that represents the conservation of energy in the system, equating the initial potential energy to the sum of the final kinetic energy and the final potential energy. Identify and define the variables in the equation based on the given problem.

- $N - 63P - 2$ Use relevant scientific principles or formulas to express the variables of interest in the problem. In this case, apply the physics of projectile motion to derive

an expression for the time of flight of a projectile, which depends on the initial speed, launch angle, and acceleration due to gravity.

- $N - 71P - 2$ Use the derived relationship from the conservation of momentum to express the velocities of the objects in terms of their masses. Substitute these expressions into the kinetic energy formula to represent the kinetic energy of each object in terms of the masses and one of the velocities.

**Plans sampled from the $3$-depth**:

- $N - 3P - 3$ Perform the necessary mathematical operations, including multiplication and division of numbers and exponents, and the square root operation, to simplify the expression and calculate the desired quantity.

- $N - 13P - 3$ Expand and simplify the terms in the equation to make it easier to isolate and solve for the unknown variable.

- $N - 19P - 3$ Perform the necessary mathematical operations to derive the final value from the previously set up equation or formula.

- $N - 25P - 3$ Rearrange the derived equation to isolate the desired variable. Substitute the known values into the equation to calculate the value of the desired variable.

- $N - 33P - 3$ Perform the necessary mathematical operations to compute the desired quantity using the formula and the values substituted into it.

- $N - 44P - 3$ Combine the derived expressions from previous steps to form an equation that can be solved for the unknown variable. This involves equating the two expressions that represent the same physical quantity, simplifying the equation by cancelling out common terms, and then rearranging the equation to solve for the unknown variable.

- $N - 52P - 3$ Simplify the derived expressions for kinetic energy by cancelling out common terms. Use the established relationships from previous steps to express all kinetic energies in terms of the same velocity and their respective masses.

**Plans sampled from the $5$-depth**:

- $N - 5P - 5$ Perform the final mathematical operation, in this case, the square root, to obtain the final result of the calculation.

- $N - 15P - 5$ Solve the simplified equation to find the value of the unknown variable, which represents the solution to the problem.

- $N - 21P - 5$ Perform the necessary mathematical operations to derive the final value from the previously set up equation or formula.

- $N - 27P - 5$ Perform the necessary calculations as per the derived equation to obtain the value of the unknown variable. This may involve multiple mathematical operations, including but not limited to, multiplication, subtraction, division, and extraction of square roots.

- $N - 35P - 5$ Perform the necessary mathematical operations to compute the desired quantity using the formula and the values substituted into it.

- $N - 46P - 5$ Simplify the mathematical expression by performing the calculations or operations indicated, and apply relevant mathematical functions or principles to derive the final value of the unknown variable.

- $N - 54P - 5$ Substitute known or given values into the simplified expression to calculate the desired quantity. Simplify the expression further if possible.

**Plans sampled from the $7$-depth**:

- $N - 7P - 7$ Evaluate the final mathematical expression to obtain the solution to the problem in the desired units.

- $N - 29P - 7$ **Verify the units of the final answer to ensure they match the expected units for the quantity being calculated. This step confirms the consistency and correctness of the solution.**

- $N - 37P - 7$ Convert the calculated results into the units specified or most appropriate for the context of the problem.

- $N - 48P - 7$ Evaluate the final mathematical expression using a calculator or appropriate software, and interpret the result in the context of the problem.

- $N - 56P - 7$ Perform the final arithmetic operation to convert the fraction into a decimal or percentage, providing a more understandable representation of the calculated quantity.

**Plans sampled from the $10$-depth**:

- Finalize the solution by evaluating the mathematical expression, ensuring that the result is in the desired units. This completes the problem-solving process.

- **Review the final results for correctness and reasonableness based on the context and known characteristics of the problem. If the results align with expectations, conclude that the problem is solved.**

E.2   SOLVED QUESTIONS

**Question:** A neutron at rest decays (breaks up) to a proton and an electron. Energy is released in the decay and appears as kinetic energy of the proton and electron. The mass of a proton is 1836 times the mass of an electron. What fraction of the total energy released goes into the kinetic energy of the proton?

*P-1: Apply the principle of conservation of momentum to establish an equation representing the relationship between the masses and velocities of the objects involved in the event.*

**S-1:** Since the neutron is initially at rest, the total momentum before the decay is zero. After the decay, the total momentum must still be zero due to the conservation of momentum. This means that the momentum of the proton and the electron must be equal in magnitude but opposite in direction. Therefore, we can write the equation for the conservation of momentum as:\n\nm_p * v_p = m_e * v_e\n\nwhere m_p and v_p are the mass and velocity of the proton, and m_e and v_e are the mass and velocity of the electron.

*P-2: Use the derived relationship from the conservation of momentum to express the velocities of the objects in terms of their masses. Substitute these expressions into the kinetic energy formula to represent the kinetic energy of each object in terms of the masses and one of the velocities.*

**S-2:** The kinetic energy of a particle is given by the equation K = 1/2 * m * v^2. We can express the velocities v_p and v_e in terms of the masses m_p and m_e using the conservation of momentum equation from Step 1. This gives us:\n\nv_p = (m_e/m_p) * v_e\n\nand\n\nv_e = (m_p/m_e) * v_p\n\nWe can substitute these expressions into the kinetic energy equations to get the kinetic energy of the proton and the electron in terms of the masses and one of the velocities. This gives us:\n\nK_p = 1/2 * m_p * ((m_e/m_p) * v_e)^2\n\nand\n\nK_e = 1/2 * m_e * ((m_p/m_e) * v_p)^2

*P-3: Simplify the derived expressions for kinetic energy by cancelling out common terms. Use the established relationships from previous steps to express all kinetic energies in terms of the same velocity and their respective masses.*

**S-3:** Simplify the expressions for the kinetic energy of the proton and the electron. The mass terms in the expressions for K_p and K_e will cancel out, giving us:\n\nK_p = 1/2 * m_e * v_e^2\n\nand\n\nK_e = 1/2 * m_p * v_p^2\n\nSince the velocities v_p and v_e are equal in magnitude (from the conservation of momentum), we can substitute v_e for v_p in the equation for K_e, giving us:\n\nK_e = 1/2 * m_p * v_e^2\n\nNow we have the kinetic energy of the proton and the electron in terms of the same velocity, v_e, and their respective masses.

*P-4: Combine the simplified expressions for the kinetic energies of the objects to represent the total energy. Calculate the fraction of the total energy that goes into one object by dividing its kinetic energy by the total energy. Simplify the expression for this fraction by cancelling out common terms.*

**S-4:** *The total energy released in the decay is the sum of the kinetic energy of the proton and the electron. We can express this as:\n\n K_total = K_p + K_e\n\nSubstituting the expressions for K_p and K_e from Step 3 into this equation gives us:\n\n K_total = 1/2 * m_e * v_e^2 + 1/2 * m_p * v_e^2\n\nWe can factor out the common terms to simplify this expression:\n\n K_total = 1/2 * v_e^2 * (m_e + m_p)\n\nThe fraction of the total energy that goes into the kinetic energy of the proton to the total energy:\n\n Fraction = K_p / K_total\n\nSubstituting the expressions for K_p and K_total into this equation gives us:\n\n Fraction = (1/2 * m_e * v_e^2) / (1/2 * v_e^2 * (m_e + m_p))\n\nThe v_e^2 terms cancel out, leaving us with:\n\n Fraction = m_e / (m_e + m_p)*

*P-5: Substitute known or given values into the simplified expression to calculate the desired quantity. Simplify the expression further if possible.*

**S-5:** We know that the mass of a proton is 1836 times the mass of an electron. We can substitute this into the equation for the fraction of the total energy that goes into the kinetic energy of the proton. This gives us:\n\nFraction = m_e / (m_e + 1836*m_e)\n\nWe can factor out m_e from the denominator to simplify this expression:\n\nFraction = 1 / (1 + 1836).

*P-6: Perform the final calculation using the simplified expression to obtain the desired result.*

**S-6:** The next step is to calculate the value of the fraction. This fraction represents the portion of the total energy that goes into the kinetic energy of the proton. The calculation gives us:\n\nFraction = 1 / (1 + 1836) = 1 / 1837.

*P-7: Perform the final arithmetic operation to convert the fraction into a decimal or percentage, providing a more understandable representation of the calculated quantity.*

**S-7:** The final solution is to simplify the fraction to get the decimal representation. This can be done by dividing 1 by 1837. The calculation gives us:\n\nFraction = 1 / 1837 ≈ 0.000544. \n\nThis means that approximately 0.0544% of the total energy released goes into the kinetic energy of the proton.

Figure 9: Illustration of how GPT-4 addresses the question guided by the plans retrieved from the plan tree of Kinetics of `TheoremQA`.

