# OpenReview forum: "$Staple$: Towards Reliable Problem Solving with Large Language Models via Plan Optimization and Tree Search"
_ICLR.cc/2025/Conference — Submitted to ICLR 2025_

### Official Review · Reviewer_NGwi · 2024-10-31

**Soundness:** 3
**Presentation:** 3
**Contribution:** 3
**Rating:** 6
**Confidence:** 4

**Summary:**

This paper proposes a step-wise plan retrieval augmented reasoning framework called Staple, which utilizes offline plan optimization. It constructs an offline plan database using tree structures via Monte Carlo Tree Search (MCTS). The plan database can be reused, updated, and expanded by users for a wider range of applications.

**Strengths:**

1. The design and construction of step-wise plan database based on tree structure and MCTS is novel.
2. The proposed framework proves to be cost-efficient and the offline database is reusable.

**Weaknesses:**

1. The tasks tested in this paper are mainly mathematics reasoning or scientific reasoning. I doubt its effectiveness in a dynamic environment where LLMs need to interact with the environment. It would be more convincing to evaluate interactive sequential decision-making benchmarks, such as ALFWorld[1], WebShop[2], WebArena[3].
2. Lack of implementation details and ablation study of reward assignment and value function. Specifically, LLM's evaluation score is implemented by a simple prompt. How will it affect the performance? Regarding value function, please provide more details and explain how it works during adaptive search.

References:

[1] ALFWorld: Aligning Text and Embodied Environments for Interactive Learning

[2] Webshop: Towards scalable real-world web interaction with grounded language agents.

[3] Webarena: A realistic web environment for building autonomous agents.

**Questions:**

1. For adaptive search, what's the search space of each reasoning step? How the progress is controlled if LLMs don't strictly follow the plan. For example, during step n, LLMs find a mistake in previous step and try to fix it.
2. The direct search method executes the plan sequence in a single interaction, instead of an iterative manner. Will this design choice influence the result?

---

### Official Review · Reviewer_6tg7 · 2024-11-02

**Soundness:** 2
**Presentation:** 1
**Contribution:** 2
**Rating:** 3
**Confidence:** 3

**Summary:**

The paper proposes "STAPLE", which constructs natural language plans using MCTS to construct a "plan database", and later uses the database to guide LLM reasoning. The paper experimented with three reasoning datasets AQUA-RAT, MATH, and TheoremQA, and the experiment results show that STAPLE achieves similar performance as other methods while requiring fewer tokens during inference.

**Strengths:**

Experimental results suggest the obtained plan database is effective and achieves comparable performance with other methods while requiring smaller number of tokens during inference.

**Weaknesses:**

- The paper suggests that doing an offline plan optimization can achieve a "once-for-all" effect where the obtained plans can be reused in future inference, however, it is not clear how costly it is for constructing such a plan database, and how transferable it is to other tasks.

- The presentation of the paper is poor, there are many unnecessary notations that are not clearly defined, and many concepts are not made clear. See Questions section for details.

**Questions:**

1. line 73, what is "node" and what is "state", I assume it's referring to "instruction" and "thought"?

2. line 175: should be argmax

3. Section 3.2  is very confusing, what is “plan”, “node”, and “thought”? I assume plan/node might be the “instruction” phi_i in section 3.1, and “thought” might be z_i in section 3.1, but it is still not clear what it is. I strongly suggest giving an example in the paper to help explain.

4. line 191-195, since \psi is generated conditioned on z_{0…n}, why can it be used as guidance for the n-th thought generation, is this referring to the later online plan searching?

5. line 206, bold \psi should be \Psi?

6. Fig. 1b is confusing, bar and curve, which is number of nodes and which is solving rate?

7. line 271-275, The difference between Direct and Adaptive is not clear to me, the paper mentions that Direct obtains a sequence of plans and LLM answers the question in a “single interaction”, since the generation alternates between plan and thoughts, how can it be “single interaction”?

8. It seems Direct is not conditioned on thoughts generated, but selects “highest r at each level”, and the reward defined in line 251 is conditioned on the thoughts, this seems contradicting.

9. I assume Ensemble-C is ensembling Direct, how about doing majority votes on Adaptive?

10. line 414, U_{800} is not defined before.

---

### Official Review · Reviewer_fmUP · 2024-11-03

**Soundness:** 2
**Presentation:** 1
**Contribution:** 3
**Rating:** 3
**Confidence:** 3

**Summary:**

The article presents "Staple", a framework designed to improve the reliability of multi-step reasoning by large language models. It addresses issues such as hallucinations and ineffective guidance by using a plan retrieval augmented reasoning system. Staple combines offline plan optimization and online plan searching to build a reusable database of high-level, question-agnostic plans. Through Monte Carlo Tree Search, the offline stage constructs and optimizes plan trees that can be reused for different problem-solving tasks, reducing token usage and interaction costs compared to traditional methods. Tested on datasets such as AQUA-RAT, MATH, and TheoremQA, Staple demonstrates competitive solving rates and efficiency.

**Strengths:**

+ Focuses on reducing token usage, an important problem for complex tasks.
+ The authors test various prompt optimization techniques to compare with their method.
+ Detailed evaluation parameters for language models are provided.
+ Code for reproducibility is included, allowing readers to replicate results.

**Weaknesses:**

Structure & Writing:

- First, the organization of content could be improved. For example, on line 156, introducing "the concept of a plan" before defining it disrupts reader comprehension. I suggest placing the definition of "plan" earlier or integrating it into the text without creating a separate "Definition" section.

- The paper would benefit from including key diagrams in the main text rather than in the appendix, particularly the "Plan Tree" referenced on line 187. For instance, it would be helpful to include a simplified version of Figure 5 or Figure 6 in the main text.

- While the authors discuss some limitations in the appendix, moving this discussion to the main body would better highlight potential challenges and areas for future research.

Data Presentation:

- The data presentation requires clarification. Table 2 does not specify whether the numbers represent success rates or utilization scores. For example, the phrase "Success rate of models across..." could be added for clarity. Additionally, there are inconsistencies in citation formatting and unnecessary repetition of algorithm names such as Monte Carlo Tree Search. For instance, on line 038, the authors use parentheses for citation, whereas on line 044, the author name and year format is used.

Methodology:

- The scope of testing appears limited. Although the paper indicates that four models would be evaluated, Figure 1 and Table 1 only show results for GPT-4. The conclusions would be more robust if the algorithm were tested beyond mathematical problems to show broader applicability. For example, including applications in program synthesis would add valuable context.

**Questions:**

Q1: What is the average cost of creating the plans database?

Q2: Would you consider providing results for other models such as Llama 2-13b?

Q3: Have you considered evaluating your methodology on one or two additional subjects (even at a smaller scale)?

---

### Official Review · Reviewer_tVhj · 2024-11-05

**Soundness:** 3
**Presentation:** 2
**Contribution:** 2
**Rating:** 3
**Confidence:** 3

**Summary:**

The paper proposes a framework addressing the limitations of traditional prompting methods that can lead to high token usage and inconsistent guidance. The method consists of two stages, the first stage is to employ a structured plan database optimized offline, using Monte Carlo Tree Search (MCTS). During online operations, the framework retrieves the optimal plans for each step, enhancing both efficiency and reliability. Empirical results demonstrate that the proposed framework achieves competitive solving rates with lower resource consumption.

**Strengths:**

1. This paper proposes an interesting and novel idea about offline optimization via MCTS and online reuse by search.
2. This paper compares various baselines with the proposed methods.
3. The authors design a complex procedure to solve the limitations of MCTS with LM, e.g. the exploration and hallucination problem.

**Weaknesses:**

1. Line 176, why using min, i.e. P* is the plan that minimizes the average accuracy.
2. A diagram about the overall framework is required.
3. An extra baseline about online MCTS (either [1][2][3]) should be added.

**Questions:**

1. Line 269, if multiple plan trees can be retrieved, will a search on multiple trees improve the results?
2. Line 269, if I want to deploy the proposed framework into other tasks that do not have category ID, how can I get the corresponding plan tree?
3. What's the computation cost of offline plan optimization?


## Reference


[1] Hao S, Gu Y, Ma H, et al. Reasoning with language model is planning with world model[J]. arXiv preprint arXiv:2305.14992, 2023.

[2] Feng X, Wan Z, Wen M, et al. Alphazero-like tree-search can guide large language model decoding and training[J]. arXiv preprint arXiv:2309.17179, 2023.

[3] Qi Z, Ma M, Xu J, et al. Mutual reasoning makes smaller llms stronger problem-solvers[J]. arXiv preprint arXiv:2408.06195, 2024.

---

### Meta-Review · Area_Chair_L5JY · 2024-12-22

**Metareview:**

This paper proposes a framework called Staple which uses plan retrieval to augment reasoning of LLMs. In the offline phase, Staple constructs a plan database of general-purpose reasoning instructions, and in the online phase to answer a new question, Staple retrieves the optima step-by-step plans from the database.

I agree with the reviewers that ideas are novel, the problem of reducing token usage is important and experiments are strong. However, the offline database creation is expensive, the scope of tasks is very limited, and the overall clarity needs to be improved.

The authors did not provide a rebuttal.

I recommend rejection.

**Additional Comments On Reviewer Discussion:**

I agree with the reviewers that ideas are novel, the problem of reducing token usage is important and experiments are strong. However, the offline database creation is expensive, the scope of tasks is very limited, and the overall clarity needs to be improved.

The authors did not provide a rebuttal.

I recommend rejection.

---

### Decision · Program_Chairs · 2025-01-22

Reject